# Ensembles provably learn equivariance through data augmentation

## Abstract

Recently, it was proved that group equivariance emerges in ensembles of neural networks as the result of full augmentation in the limit of infinitely wide neural networks (neural tangent kernel limit). In this paper, we extend this result significantly. We provide a proof that this emergence does not depend on the neural tangent kernel limit at all. We also consider stochastic settings, and furthermore general architectures. For the latter, we provide a simple sufficient condition on the relation between the architecture and the action of the group for our results to hold. We validate our findings through simple numeric experiments.

## 1 Introduction

Consider a learning task with an inherent symmetry. As an illustrative example, we can think of classifying images of, say, apples and pears. The classification should of course not change when the image is rotated, meaning that the learning task is *invariant* to rotations. If we are segmenting the image, the apple segmentation mask should instead rotate with the image – such tasks are called *equivariant*. An a-priori known symmetry like this is a strong inductive bias, that should be used to improve performance of the model.

There are essentially two 'meta-approaches' to do the latter. The first one is to use data augmentation to generate new data that respects the symmetry – in our example, to add all rotated versions of the training images to the dataset. This is versatile, simple and effective. However, there are of course no guarantees that the model after training exactly obeys the symmetry, in particular on out-of distribution data. To solve this problem, one can use the second meta-approach, namely to use a model that inherently respects the symmetry. Over the last half decade or so, an entire framework concerning achieving this for neural network has emerged under the name of *Geometric Deep Learning* (Bronstein et al., 2021).

The relations between these two approaches are, from a theoretical point of view, still unclear: there is yet to emerge a definitive answer to the question if and when the augmentation strategy is guaranteed to yield an equivariant model after training. Recently, an interesting discovery in this regard was made in Gerken & Kessel (2024). In this paper, it was shown that *ensembles* of *infinitely wide* neural networks (in the neural tangent kernel limit Jacot et al. (2018)) are equivariant *in mean*, even though individual members of them are not. See Figure 1 for an illustration. Although this asymptotic result is already interesting, it has some drawbacks. The proofs rely heavily on the simplified dynamics of the NTKs, and therefore do not work at all for network of finite width. Furthermore, the proof only works for finite groups, and is only formulated for the somewhat unrealistic gradient flow training.

The main message of this paper is that the phenomenon of ensembles of neural networks being equivariant in mean is *much more general* than Gerken & Kessel (2024) suggests. Our analysis reveals that the equivariance has nothing to do with the neural tangent kernel limit. Instead, if the geometry of the neural network architecture in a sense is not inherently biased against equivariance (the technical condition is described in Definition 1), and the parameters of the ensemble are initialized in a symmetric way (which can be achieved by Gaussian initialization), the ensemble mean will automatically be equivariant at all points in the training.

Our proofs are non-asymptotic, work for any compact group, for stochastic gradient descent and random augmentation, and apply to a wide range of neural network architectures. Consequently, the

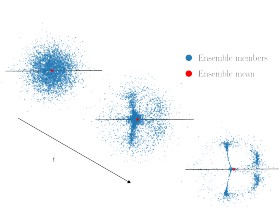

Figure 1: A graphical explaination of our results. In the example, the symmetry group is $C_2$, acting on the parameter space through reflection in the $x$-axis (this is the representation $\rho$ we define below). Parameters on the $x$-axis correspond to equivariant models. We are shown snapshots of the parameters of ensemble members as they are trained on symmetric data. Note that at all times, most individual ensemble members that do not lie on the line of symmetric model. However, since they have been initialized with a symmetric distribution, their distribution stays symmetric throughout training – and therefore, *their mean* always corresponds to an equivariant model.

analysis is different to the one in Gerken & Kessel (2024). We will in particular use recent results from Nordenfors et al. (2024) about the dynamics of neural networks trained on symmetric data.

The paper outline is as follows: After a short literature review in Section 2, we need to spend considerable time to properly present the framework and state our assumptions in Section 3. Then, we state our main result, Theorem 4.2 in Section 4, and prove it in a simplified setting – the full proof is given in the appendix. Finally, we perform some simple numerical experiments to showcase our theory in Section 5.

## 2 LITERATURE REVIEW

To give a comprehensive overview of Geometric Deep Learning goes well beyond the scope of this paper. We instead refer to the textbook (Bronstein et al., 2021).

The question of the effect of symmetries in the data on the training of neural networks has often been understood as a comparison question: Is it better to train a non-equivariant architecture on augmented (that is, artificially symmetric) data, or to use a manifestly equivariant architecture as e.g. Cohen et al. (2019); Maron et al. (2019); Kondor & Trivedi (2018); Weiler & Cesa (2019); Finzi et al. (2021); Fuchs et al. (2020)? Empirical comparisons between the two approaches as part of experimental evaluations of manifestly equivariant architectures have been made more or less since the invention of equivariant networks – more systematic investigations include Gandikota et al. (2021); Gerken et al. (2022).

In Chen et al. (2020), a group theoretical framework for data augmentation was developed, which has inspired many treatises of the question, including ours. Linear (i.e. kernel) models are studied in Elesedy & Zaidi (2021); Mei et al. (2021); Dao et al. (2019). Lyle et al. (2019; 2020) study so-called *feature-averaged* networks. So-called *linear neural networks* – i.e., neural networks with a linear activation function – are studied in Chen & Zhu (2024). In these settings, equivalence between augmenting the data and restricting the networks can be proven. The same has not been established for 'bona-fide' neural networks with non-linear activation functions – they are studied Nordenfors et al. (2024), but there, only local guarantees are derived.

The work most tightly related to ours is Gerken & Kessel (2024). In there, it was realized that equivariance emerges from augmentation in ensembles, and not in individual networks. They provide a formal explanation of this only in the so-called *neural tangent kernel (NTK) limit* (Jacot et al., 2018). Importantly, the optimization dynamics of neural networks in the NTK-limit turn linear (Lee et al., 2019), so that that setting is considerably simpler than the one we consider here.

Equivariant flows have been studied before in the context of generative models (Köhler et al., 2020; Katsman et al., 2021; Satorras et al., 2021). In this paper, we use results from Köhler et al. (2020) in a different context, namely to study the flow of the *parameters of the network* under (stochastic) gradient flow (descent).

## 3 PRELIMINARIES

### 3.1 PAPER SETTING

By and large, we adopt the framework from Nordenfors et al. (2024) – which in large is just a particular formalization of the general geometric deep learning framework Bronstein et al. (2021). Let $(x, y) \in X \times Y$, where $X$ and $Y$ are (finite-dimensional) vector spaces. We are concerned with neural networks $\Phi_A : X \to Y$, that are defined as function of the form

$$x_0 \in X, \quad x_{i+1} = \sigma_i(A_i(x_i)), \, \forall i \in \{0, 1, \ldots, N-1\}, \Phi_A(x_0) = x_N \in Y. \tag{1}$$

Here, $A_i$ are trainable linear maps, i.e. elements of $\operatorname{Hom}(X_i, X_{i+1})$, and where $\sigma_i$ are fixed non-linearities. Following Nordenfors et al. (2024), we write $\mathcal{H} = \bigoplus_{i=0}^{N-1} \operatorname{Hom}(X_i, X_{i+1})$ for the ambient space for the parameter $A = (A_i)_{i=0}^{N-1}$. This is a faithful model for *fully connected* networks *without bias*. As was realized in Nordenfors et al. (2024), we may include other architectures in the framework (biases, CNN:s, RNN:s, transformers, ... ) by confining the linear maps to an *affine subspace* $\mathcal{L} \subseteq \mathcal{H}$. Let us present one example of this construction, that we in the following will use as a running example.

*Example* 3.1. For some integers $d_i$, let $X_i = (\mathbb{R}^{N,N})^{d_i}$ – we can and will interpret this as *tuples of images*. We may define a *convolutional operator* between two such spaces as operators $C$ of the form

$$(Cx)_k = \sum_{\ell \in [d_i]} \psi_{k\ell} * x_\ell,$$

where $\psi_{k\ell}$, $k \in [d_{i+1}], \ell \in [d_i]$ are filters. In the following, we denote this operator $\langle \psi \rangle$. The space of all operators of this form, where the filters supports are all contained in a fixed set $\Omega$, form a linear subspace of $\mathcal{H}$. In particular, they define an affine space $\mathcal{L}$, and the corresponding architecture is a CNN.

Common choices for $\Omega$ are $3 \times 3$ or $5 \times 5$ squares. Just as was done in Nordenfors et al. (2024), we will here mainly consider the two non-canonical supports in Figure 2, since they illustrate important aspects of our results. We call the left support the *symmetric* support, and the right the *asymmetric* support, and denote the corresponding affine spaces $\mathcal{L}^{\mathrm{sym}}$ and $\mathcal{L}^{\mathrm{as}}$.

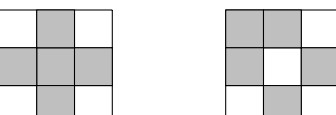

Figure 2: Left: symmetric filter. Right: asymmetric filter. Grey indices correspond to indices where the filter is supported.

Let us now bring equivariance in to the picture. We assume that a group $G$ is acting on both input, intermediate and output spaces by some *unitary representations* $\rho_i$ (Fulton & Harris, 2004) – i.e. maps $\rho_i$ from the group $G$ to the set of unitary matrices that respect the group action: $\rho_i(gh) = \rho_i(g)\rho_i(h)$. The representations on the spaces $X_i$ yield representations $\widehat{\rho}_i$ on $\operatorname{Hom}(X_i, X_{i+1})$ through $\widehat{\rho}_i(g)(A_i) = \rho_{i+1}(g) \circ A_i \circ \rho_i(g)^{-1}$, which in turn defines a representation $\rho$ on $\mathcal{H}$ by taking the direct product of the $\widehat{\rho}_i$. We will refer to these as the *lifted representations*.

*Example* 3.2. The group $C_4$, consisting of the four rotations in the plane of multiples of $\pi/2$, naturally acts on tuples of images by rotating each image in the tuple. We will denote this representation $\rho^{\mathrm{rot}}$. The corresponding lifted representation has an important property in relation to the convolutional operators as in Example 3.1: it acts by rotating each filter, $\rho(g)\langle \psi \rangle = \langle \rho^{\mathrm{rot}}(g)\psi \rangle$. This is not hard to show – a proof can for instance be found in Nordenfors et al. (2024).

*Remark* 1. While the representations on the input and output space are fixed – they are given by the symmetries of the problem – the representations on the intermediate spaces are a priori free to choose. This point has subtle but important consequences for the meaning of our main results – we will discuss them when we present it.

We call a map $f : X \to Y$ *equivariant* if $f \circ \rho_0(g) = \rho_N(g) \circ f$ for all $g$. The space of $G$-equivariant linear maps between $X_i$ and $X_{i+1}$ is denoted $\operatorname{Hom}_G(X_i, X_{i+1})$. We write $\mathcal{H}_G = \bigoplus \operatorname{Hom}_G(X_i, X_{i+1})$, i.e., $\mathcal{H}_G = \{A \in \mathcal{H} : \rho(g)A = A\}$.

Now let us record and discuss three global assumptions we want to make, which are similar to the ones made in Nordenfors et al. (2024).

**Assumption 1.** *The non-linearities $\sigma_i$ are all $G$-equivariant.*

**Assumption 2.** *The group $G$ is compact.*

**Assumption 3.** *The space $\mathcal{H}_G \cap \mathcal{L}$ is nonempty.*

As for Assumption 1, note that without it, the common strategy to define equivariant neural networks by restricting the linear layers to lie in this space will not work. Assumption 2 is needed to ensure that the Haar measure of the group (Krantz & Parks, 2008) is (normalizeable to) a probability measure. It should also be noted that when $G$ is compact, it is no true restriction to assume that the representation is unitary – one can always ensure this via redefining the inner products on $\mathcal{H}$. The assumption is satisfied by all finite groups, but also groups like $\mathrm{SO}(n)$ and $\mathrm{O}(n)$ (albeit not the group of all rigid motions $\mathrm{SE}(n)$.) As for Assumption 3, it ensures that there exists an admissible and $G$-equivariant architecture. It also has a technical consequence that we will need: it lets us write the affine space $\mathcal{L} = A_{\mathcal{L}} + \mathrm{T}\mathcal{L}$, for an $A_{\mathcal{L}} \in \mathcal{H}_G \cap \mathcal{L}$, and $\mathrm{T}\mathcal{L}$ the direction of $\mathcal{L}$.

### 3.2 TRAINING ALGORITHMS

Let us now assume that we are given data $(x, y)$ distributed according to some distribution on $X \times Y$. We let $\ell : Y \times Y \to \mathbb{R}$ be an invariant loss function ($\ell(\rho_N(g)y, \rho_N(g)y') = \ell(y, y')$ for all $g \in G$ and $y, y' \in Y$) , and define a nominal risk through

$$R(A) = \mathbb{E}_{\mathcal{D}}(\ell(\Phi_A(x), y)).$$

We can then train networks via minimizing $A$. Since $A$ is confined to the affine space $\mathcal{L}$, we thereby technically do not change $A$ by applying e.g. gradient descent to $R$ directly – instead, we parametrise $A = A_{\mathcal{L}} + Lc$, where $A_{\mathcal{L}} \in \mathcal{L}$ and $L : \mathbb{R}^p \to \mathrm{T}\mathcal{L}$ is a unitary ($L^*L = \mathrm{id}$) parametrization of the direction of $\mathcal{L}$, and then update $c$ via minimizing the loss $\mathrm{R}(c) = R(A_{\mathcal{L}} + Lc)$.

If the distribution $\mathcal{D}$ is biased, in the sense that $(\rho(g)x, \rho(g)y)$ is not equally distributed to $(x, y)$, training the network with this risk will most likely not lead to an equivarant model. Instead, we need to augment the data. We consider two strategies to do so.

**(a) Gradient flow with full augmentation.** *Full augmentation* refers to training the networks using data distributed according to $(\rho(g)x, \rho(g)y)$, $(x, y) \sim \mathcal{D}$ and $g$ drawn according to the *Haar measure* Krantz & Parks (2008). For a finite group, this corresponds to drawing an element uniformly at random – but it can also be defined on any *compact* group, such as $\mathrm{SO}(n)$, as the unique *invariant* probability measure on $G$ – i.e., if $g \sim \mu$, $gh \sim \mu$ as well for any $h \in G$. This change can mathematically be expressed as changing the loss function to the augmented loss

$$R^{\mathrm{aug}}(A) = \int_G \mathbb{E}_{\mathcal{D}}(\ell(\Phi_A(\rho_0(g)x), \rho_N(g)y))\mathrm{d}\mu(g),$$

and similarly $\mathrm{R}^{\mathrm{aug}}(c) = R^{\mathrm{aug}}(A_{\mathcal{L}} + Lc)$. We consider the simplified setting of using an infinitesemal learning rate, e.g. training the network using *gradient flow*. The latter refers to let the parameters flow accoarding to $\dot{c}^t = -\nabla \mathrm{R}^{\mathrm{aug}}(c^t)$. Note that due to the chain rule and the unitarity of $L$, $A^t = A_{\mathcal{L}} + Lc^t$ will then follow the *projected gradient flow*

$$\dot{A}^t = -\Pi_{\mathcal{L}}\nabla R^{\mathrm{aug}}(A^t), \tag{2}$$

where $\Pi_{\mathcal{L}} = LL^*$ is the orthogonal projection onto $\mathrm{T}\mathcal{L}$, the direction of $\mathcal{L}$. The differential equation equation 2 induces a flow map $\mathcal{F}^t : \mathcal{L} \to \mathcal{L}$: Every $A \in \mathcal{L}$ is mapped to the value at time $t$ of the solution of equation 2 initialized at $A$.

**(b) SGD with random augmentation.** In addition to the above model, we will also consider training the networks on randomly augmented minibatches. This both escapes the assumption of full augmentation (which is impractical for large, and impossible for infinite, groups) and includes finite learning rates.

Just to fix notation, let us define the procedure formally. We define the *partially augmented and sampled risk* as

$$R^g(A) = \frac{1}{s}\sum_{k=1}^{s} \ell(\Phi_A(\rho_0(g_k)x_k), \rho_N(g_k)y_k),$$

where $(x_k, y_k) \sim \mathcal{D}$ are independent, both from each other as well as from the $g_k$, which are drawn i.i.d according to $\mu$. Note that $R^g$ is a random function. The sample size $s$ is a parameter that is free to chose.

Given a sequence of learning rates $(\gamma_t)_{t \geq 0}$, we may now iteratively define a sequence of random variables as follows

$$A^{t+1} = A^t - \gamma_t \Pi_{\mathcal{L}} \nabla R^g(A^t).$$

Note that the projected descent emerges in the same way as above via updating the parametrizing coefficients $c^t$. Every realization of the process $A^t$ corresponds to one run of the SGD.

### 3.3 ENSEMBLES

We define ensembles as in Gerken & Kessel (2024): For a distribution $p$ on $\mathcal{H}$, we consider the distribution of parameters $A^t$ implied by initalizing the parameters of a network according to $p$, and then training the network for time $t$. The corresponding *ensemble model* at time $t$ is then defined as

$$\overline{\Phi}_t(x) = \mathbb{E}_\pi[\Phi_{A^t}(x)], \tag{3}$$

i.e. the expecation over $\pi$, which denotes $p$ in the deterministic gradient flow setting, and the product of $p$ with the random draw of the minibatches and group elements in the SGD setting. Note that this function can be well approximated by drawing a sample mean, i.e. starting $N$ neural networks with i.i.d. initialisation (drawn according to $p$), training them, and then calculating the mean after $t$ points. This is the strategy we will follow in the experimental section. We will however only make claims about the true mean, and leave the study of the effect of the approximation error to further work. We will refer to such collections of $N$ networks as an *ensemble*, and each individual network an *ensemble member*.

## 4 THEORY

This section is devoted to proving the $G$-equivariance of ensembles (Theorem 4.2). Our result will hold under a geometric condition on the space $\mathcal{L}$. Let us present it and discuss it in some detail.

### 4.1 THE $G$-INVARIANCE CONDITION

**Definition 1.** *We say that $\mathcal{L}$ is $G$-invariant if $\rho(g)\mathcal{L} \subseteq \mathcal{L}$ for all $g \in G$.*

It should be noted that the invariance of $\mathcal{L}$ depends on the representation $\rho$, and not just on $G$. However, we find this terminology more intuitive, and we will make sure to point out the dependence when it matters.

From the view of the neural network architectures, $G$-invariance can be interpreted as saying that there is no *inherent bias* against being $G$-equivariant in the definition of your admissible layers. It has the for us important consequence, which we prove in Appendix A.

**Lemma 4.1.** *$\mathcal{L}$ is $G$-invariant if and only if $\Pi_{\mathcal{L}}$ is $G$-equivariant.*

$G$-invariance is related to the so-called *compatibility condition* that was used in Nordenfors et al. (2024) to prove statements about the stationary points of the augmented risk $R^{\text{aug}}$: that the projection $\Pi_{\mathcal{L}}$ commutes with the orthogonal projection $\Pi_G$. In fact, the $G$-invariance was there argued to be sufficient for the compatibility condition to hold. It was furthermore argued that $G$-invariance is satisfied for essentially all commonly used architectures and reasonable representations, such as fully connected ones with and without bias, RNNs, residual connections, etc. Let us here, again, only discuss our running examples.

*Example* 4.1. In Example 3.1, it was argued that when using the canonical $\rho^{\text{rot}}$-representations on all the intermediate layers, we have $\rho(g)\langle\psi\rangle = \langle\rho(g)\psi\rangle$. Consequently, a $\langle\psi\rangle \in \mathcal{L}^{\text{sym}}$ will be mapped by $\rho$ onto a tuple of other symmetric filters, and hence $\mathcal{L}^{\text{sym}}$ is $G$-invariant. By the same argument, $\mathcal{L}^{\text{as}}$ is not.

As we previously remarked (Remark 1), there is no need to choose the $\rho^{\text{rot}}$-representations in the intermediate layers. However, it turns out that there aren't *any* representations on the intermediate spaces which makes $\mathcal{L}^{\text{as}}$ $G$-invariant. The somewhat involved proof for this is given in Appendix C.

## 4.2 MAIN RESULT

Let us now state our main result.

**Theorem 4.2.** *If (i) the parameters have a $G$-invariant distribution at initialization, $\rho(g)A^0 \sim A^0$ (ii) the affine space $\mathcal{L}$ of the architecture is $G$-invariant, then the ensemble model trained with either gradient flow with full augmentation or SGD with random augmentation will for all times $t$ be $G$-equivariant.*

*Remark* 2. It is not hard fulfill condition (i) if $\mathcal{L}$ is $G$-invariant – simply choose the $c$-parameter as a standard Gaussian vector. Then, $A = A_{\mathcal{L}} + Lc$ will also be a Gaussian vector on the space $\mathcal{L}$, invariant to all the unitary transformations $\rho(g)$. A detailed argument is given in Appendix A.1.

*Remark* 3. As we remarked in Remark 1, the representations on the intermediate spaces $X_i$ are free to choose. One should hence really interpret condition (ii) in Theorem 4.2 as that there *exist* intermediate representations so that $\mathcal{L}$ is $G$-invariant with respect to the corresponding $\rho$.

We will here only prove Theorem 4.2 for the case of gradient flow. The proof in the stochastic case in fact is essentially the same, but needs a little more tools from probability theory, whence we postpone it to Appendix B.

To show that $\overline{\Phi}_t(x)$ is equivariant, it is enough to prove an invariance result of the *distribution* of the parameters $A^t$, as the following lemma shows.

**Lemma 4.3.** *If the distribution of $A^t$ on $\mathcal{H}$ is group invariant in the sense that $\rho(g)A^t \sim A^t$ for all $g \in G$, then $\overline{\Phi}_t$ is $G$-equivariant.*

*Proof.* In the proof of Lemma 3.9 in Nordenfors et al. (2024), it is shown that $\Phi_A(\rho_0(g)x) = \rho_N(g)\Phi_{\rho(g^{-1})A}(x)$, for $g \in G$ and $x \in X$ – we give the relevant part of the proof in Appendix A. This formula allows us to argue that if $A^t \sim \rho(g)A^t$, $\forall g \in G$, we have for any $x$

$$\Phi_{A^t}(\rho_0(g)x) = \rho_N(g)\Phi_{\rho(g^{-1})A^t}(x) \sim \rho_N(g)\Phi_{A^t}(x), \quad \forall g \in G.$$

Taking the expectation over $\pi$ now gives the claim. $\square$

We thus have the clear goal of finding reasonable assumptions under which the condition of Lemma 4.3 is fulfilled. Remember that gradient flow amounts to applying the flow map $\mathcal{F}^t$ to the parameters of each ensemle member. If that flow map is equivariant, it will keep an invariant initial distribution of $A$ invariant – this is the content of e.g. Theorem 1 from Köhler et al. (2020). Let us formulate this as a lemma.

**Lemma 4.4.** *If $\rho(g)A \sim A$, $\forall g \in G$, and $\mathcal{F}^t$ is $G$-equivariant, then $\rho(g)\mathcal{F}^t A \sim \mathcal{F}^t A$, $\forall g \in G$.*

*Proof.* Note that if $\rho(g)A \sim A$, $\forall g \in G$, then $\mathcal{F}^t \rho(g)A \sim \mathcal{F}^t A$, since $\mathcal{F}^t$ and $\rho(g)$ are Borel measureable, $\forall g \in G$. If we now assume that $\mathcal{F}^t$ is $G$-equivariant, then $\rho(g)\mathcal{F}^t A = \mathcal{F}^t \rho(g)A \sim \mathcal{F}^t A$, $\forall g \in G$. $\square$

This means that we want to look for assumptions that guarantee $\mathcal{F}^t$ is $G$-equivariant. Well, in fact, Köhler et al. (2020) showed that the flow map given by a $G$-equivariant vector field is also $G$-equivariant, the proof of which we include in Appendix A.

**Lemma 4.5.** *The flow generated by a $G$-equivariant vector field is $G$-equivariant.*

This means for us that we only have left to show that $-\Pi_{\mathcal{L}}\nabla R^{\mathrm{aug}}$ is $G$-equivariant, which is the content of the following lemma.

**Lemma 4.6.** *If $\mathcal{L}$ is $G$-invariant, then $-\Pi_{\mathcal{L}}\nabla R^{\mathrm{aug}}$ is $G$-equivariant.*

*Proof.* Let $\mathcal{L}$ be $G$-invariant. Then by Lemma 4.1 it follows that $\Pi_{\mathcal{L}}$ is $G$-equivariant. Thus, since the composition of $G$-equivariant maps is again $G$-equivariant, we only need to show that $\nabla R^{\mathrm{aug}}$ is $G$-equivariant. To this end it suffices to show that $R^{\mathrm{aug}}$ is $G$-invariant, since the $G$-equivariance of the gradient then follows by the chain rule. To show this, we will use Lemma 3.9 from Nordenfors et al. (2024), which states that

$$R^{\mathrm{aug}}(A) = \int_G R(\rho(g)A)\mathrm{d}\mu(g).$$

Now the $G$-invariance of $R^{\mathrm{aug}}$ is a straightforward matter. Utilizing that $\rho$ is representation and the invariance of the Haar measure, we obtain for every $h \in G$

$$R^{\mathrm{aug}}(\rho(h)A) = \int_G R(\rho(gh)A)\mathrm{d}\mu(g) = \int_G R(\rho(g)A)\mathrm{d}\mu(g) = R^{\mathrm{aug}}(A),$$

This concludes the proof. $\qquad\square$

Now, let us put all of this together to prove Theorem 4.2.

*Proof of Theorem 4.2.* Assume that $\rho(g)A^0 \sim A^0$ and that $\mathcal{L}$ is $G$-invariant. By Lemma 4.6 we have that $-\Pi_{\mathcal{L}}\nabla R^{\mathrm{aug}}$ is $G$-equivariant. Then Lemma 4.5 yields that $\mathcal{F}_t$ is $G$-equivariant, from which it follows by Lemma 4.4 that $\rho(g)\mathcal{F}_t A^0 \sim \mathcal{F}_t A^0$, $\forall g \in G$. Finally, it follows by Lemma 4.3 that $\overline{\Phi}_t$ is $G$-equivariant, since $\mathcal{F}_t A^0 = A^t$, which completes the proof. $\qquad\square$

This concludes the proof of Theorem 4.2 in the case of gradient flow and full augmentation.

*Remark* 4. As we have earlier noted, full group augmentation is practically impossible when dealing with infinite groups. Theorem 4.2 is however still relevant for it. Namely, we could approximate the full augmentations by just augmenting with finite subset $H$ of $G$. Then, the risk we should optimize is $R^H(A) = \frac{1}{|H|}\sum_{h \in H} R(\rho(h)A)$. If $H$ is a subgroup of $G$, this directly corresponds to performing full augmentation with the smaller set $H$. For instance, one could approximate the group of rotations $\mathrm{SO}(3)$ by the subgroup of order 60 of rotational symmetries of an icosahedron. Our results then immediately imply that the ensemble models will be equivariant with respect to that subgroup.

As stated earlier, in Appendix B we prove that Theorem 4.2 also holds for SGD with random augmentation. The proof follows roughly the same idea as here: First, we show that the gradient field $\nabla R^g$ in a sense is equivariant. Importantly, it does not hold in the sense that $\nabla R^g$ for a fixed draw of $g$ is an equivariant vector field, but rather that $\nabla R^g(\rho(g)A) \sim \rho(g)R^g(A)$, for every $g$ and $A$ – i.e., that there holds an equivariance *in distribution*. We then show that the 'flow' of the SGD with respect to such an equivariant vector field will be equivariant, and thus transform $G$-invariantly distributed parameters $A^0$ to $G$-invariantly distributed parameters $A^t$ after $t$ steps.

## 5 EXPERIMENTS

We perform two small numerical experiments, to test the robustness of our theorems to using the sample mean instead of the true expectation when creating the ensemble model and the necessity of the assumptions that $\mathcal{L}$ is $G$-invariant and that the initial distribution of parameters is $G$-invariant. The code, that relies heavily of the code made available from Nordenfors et al. (2024), can be found in the supplementary materials, and additional technical details (such as specification of hardware) in Appendix D.

We use two metrics to evaluate the invariance of the trained models. First, similarly as in Gerken & Kessel (2024), we calculate the average *orbit same prediction* (OSP). That is, for each test image, we test how many of the transformed versions of it are given the same prediction as the untransformed one by the ensemble model. OSP lies between 1 and $|G|$, where a value equal to the order of the group $|G|$. indicates perfect invariance. The second metric we calculate is the average symmetric KL-divergence $D_{\mathrm{KL}}$ between the class probabilities predicted on an image with each of the transformed versions of it. Note that the class probabilities can vary quite a lot but still yield the same prediction, whence this is a finer measure of invariance than the OSP. Both metrics are evaluated both on the MNIST test set, as well the (out-of-distribution) CIFAR-10 (Krizhevsky, 2009) test set (reshaped and converted to grayscale).

### 5.1 EXPERIMENT 1: $C_4$

We train ensembles of 1000 small CNNs of the same structure that is used in Nordenfors et al. (2024) (see Figure 3 as well as Appendix D). We train nearly identical ensembles, the only difference being that the support of the filters: the members of one ensemble have filters of symmetric support, whereas those of the other have asymmetric support. The models are trained for 10 epochs on the

$\rho^{\text{rot}}$
$\rho^{\text{triv}}$

$$X \xrightarrow[\text{tanh, LayerNorm}]{\text{Conv, Pool,}} X_1 \xrightarrow[\text{tanh, LayerNorm}]{\text{Conv, Pool,}} X_2 \xrightarrow[\text{tanh, LayerNorm}]{\text{Conv,}} X_3 \xrightarrow[\text{Fully-Connected}]{\text{Flatten,}} Y$$

$\mathbb{R}^{1 \times 28 \times 28}$ $\quad\quad$ $\mathbb{R}^{16 \times 14 \times 14}$ $\quad\quad$ $\mathbb{R}^{16 \times 7 \times 7}$ $\quad\quad$ $\mathbb{R}^{16 \times 7 \times 7}$ $\quad\quad$ $\mathbb{R}^{10}$

Figure 3: The architecture used in our neural networks. The convolutions have filters with support as in Figure 2 (left or right).

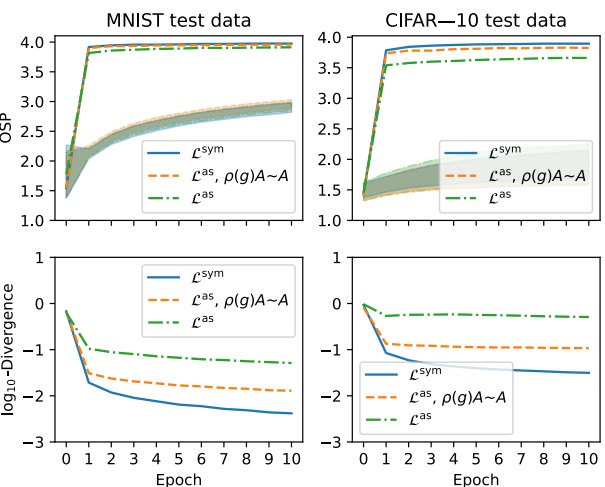

Figure 4: Top: OSP for ensembles with 1000 members.
(Higher is better)
The middle 95% of individual ensemble members are within the shaded area.
Bottom: Logarithm of symmetric Kullback–Leibler Divergence for ensembles with 1000 members.
(Lower is better).
Best viewed in color.

MNIST dataset (LeCun et al., 1998), using SGD with a constant learning rate of 0.01, a batch size of 32, and cross entropy loss. Applying augmentations using $C_4$-rotations randomly, as described in Section 3.2. Since $\mathcal{L}^{\text{sym}}$ for this representation is $G$-invariant, but $\mathcal{L}^{\text{as}}$ is not, we expect that the former ensembles become equivariant, whereas the latter do not.

We initialize the ensembles by drawing the coefficients $c$ from a standard normal distribution, as in Remark 2. Since $\mathcal{L}^{\text{as}}$ is not $G$-invariant, this does not yield an invariant initial distribution. We can mitigate this by setting the 'corner pieces' of all asymmetric filters equal to zero. We hence get three ensembles: One symmetric, one asymmetric initialized $G$-invariantly, and one asymmetric initialized "naïvely".

**Results and discussion** The evolution of the metrics along the training are shown in Figure 4. We see that with respect to both metrics, the symmetric ensembles outperform the asymmetric ones initialized invariantly, which in themselves outperform the naïvely initialized asymmetric ones. The differences are more prominent on the OOD data, which is to be expected – the networks have actually 'seen' rotated nines in the MNIST data and can hence have learned to predict them correctly even if the network is not inherently equivariant – the same is not true for rotated cars in the CIFAR10 set. We also give the metrics after epoch 10 in Table 5.1

|  | MNIST | | CIFAR–10 | |
|---|---|---|---|---|
| Model | OSP | $\log D_{\text{KL}}$ | OSP | $\log D_{\text{KL}}$ |
| $\mathcal{L}^{\text{sym}}$ | **3.97** | **−2.38** | **3.89** | **−1.50** |
| $\mathcal{L}^{\text{as}}, \rho(g)A \sim A$ | 3.96 | −1.89 | 3.82 | −0.97 |
| $\mathcal{L}^{\text{as}}$ | 3.91 | −1.29 | 3.66 | −0.29 |

Table 1: Metrics after the 10th epoch of training for full ensembles.

One should acknowledge that clearly, the ensembles, even the symmetric ones, are never truly invariant, in particular at initialization. This is not what Theorem 4.2 predicts – but note that it only makes a claim about ensemble models defined as the true expectations over the distribution of pa-

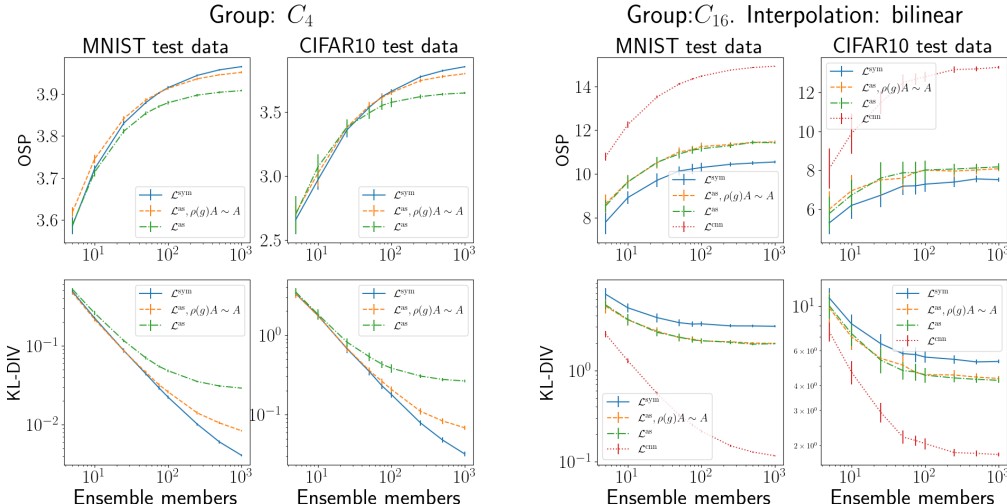

Figure 5: Metrics after the $10^{\text{th}}$ epoch for different ensemble sizes for the $C_4$ experiment (left) and $C_{16}$ experiment (right). Each datapoint is a mean of 30 bootstrapped examples – the errorbars denotes one standard deviation of the bootstrap. The $x$-scale in the top plots are logarithmic, both scales are logarithmic in the bottom plots. Best viewed in color.

rameters having the invariant property, whereas the ensembles we train here are finite. Note that it is natural that this effect is especially prominent at initialization, since the networks at initialization are entirely random, and small perturbations could yield different predictions. In fact, in our experiments the OSP at initialization does not increase as ensemble size increases. However, the more fine-grained $\log D_{\text{KL}}$-divergence measure decreases: It equals $1.13$ for a random 10-member ensemble, $0.58$ for a random 100-member ensemble, and $-0.17$ for the full ensemble. This supports that the non-perfect invariance of the ensembles can be mainly attributed to using a finite ensemble.

In Figure 5 (left), we make a more thorough investigation of the effect of using finite ensembles. Shown are the value of the metrics for randomly chosen subensembles of different sizes at the final epoch. We see that all ensembles become more equivariant as the ensemble size grows. We note that even ensembles of such a moderate size as 50 have an OSP on out-of-distribution data of above 3.5. We also see that the symmetric ensembles outperform the asymmetric ones already for quite moderate ensemble sizes. More details about the results are given in Appendix G.

One should also note that although the asymmetric ensembles are less invariant than the symmetric ones, they are still remarkably close to being invariant. This cannot be completely explained by our theory. Note that we assume $G$-invariance of $\mathcal{L}$ is order to achieve that $\rho(g)\Pi_{\mathcal{L}} = \Pi_{\mathcal{L}}\rho(g)$. However, in our case, $\rho(g)\Pi_{\mathcal{L}}$ is still "almost" equal to $\Pi_{\mathcal{L}}\rho(g)$ for the asymetric filters – it only differs for the non-zero corners. We investigate this point further in Appendix F. Our results there support this hypothesis somewhat, but more work is needed in the future.

Now let us finally comment on the fact that even the *non-invariantly initialized* ensembles seem to become more equivariant after training. Note that the main result of our paper can be interpreted as a 'stationarity' result – if we start invariantly distributed, we stay that way. Our experiment indicates that the the invariant distributions even are *attractors*. If this is true, and if so how generally, is an interesting direction of future work.

## 5.2 EXPERIMENT 2: $C_{16}$

For the second experiment, we change the symmetry group to $C_{16}$, i.e., the group of rotations of multiples of 22.5 degrees. Since the standard pixel grid is not invariant to all such rotations, this setting requires us to use interpolation. This causes the assumptions of our main result to be violated in multiple ways. For one, the maps $\rho_i : C_{16} \to \text{L}(X_i, X_{i+1})$ are formally no longer representations. They also do not act by permuting pixels, so that the lifted representations no longer act directly on

the filter (see Example 3.1 ). Even ignoring these aspects, both spaces $\mathcal{L}^{\mathrm{sym}}$ and $\mathcal{L}^{\mathrm{asym}}$ are far away from being invariant – rotating a symmetric filter by 45 degrees will send it to a filter supported on the corners of the $3 \times 3$ square. We therefore include a fourth model in these experiments, namely a standard, full $3 \times 3$-support CNN. The $\mathcal{L}^{\mathrm{cnn}}$ space is approximately invariant to $C_{16}$, and should therefore produce more equivariant models.

For these experiments, the batch size was put equal to 128, to speed up the training. Also, due to technical issues, not all ensemble members finished training – 196 $\mathcal{L}^{\mathrm{sym}}$, 151 $\mathcal{L}^{\mathrm{asym}}$ with symmetric initialization and 120 $\mathcal{L}^{\mathrm{asym}}$ with asymmetric initialization failed. The results presented below are bootstrapped ensembles from the ones that finished training.

*Remark* 5. There are different ways choosing the interpolation. Here, we will present results for a `BILINEAR` interpolation (as in the `torchvision` package) – in Appendix E, we present result also for the `NEAREST` interpolation option.

**Results and discussion**   The performance with respect to the different metrics for the four models at epoch 10 is presented in Figure 5 (right). More details are again given in in the appendix. In comparison to the the the $C_4$ experiment, all models fare much worse – notice here for instance that no model comes close to the optimal OSP of 16. Note that is what to be expected from our theory – the compatibility assumption is not fulfilled for any model. We also see that the standard CNN:s outperform the other models vastly – which is also what could be expected, given that the $3 \times 3$-square is closer to being an invariant support than are the symmetric and asymmetric supports.

# 6   CONCLUSION

The goal of this paper was to extend the results of Gerken & Kessel (2024), regarding $G$-equivariance of ensembles of infinitely wide neural networks trained with gradient descent under full augmentation, to ensembles of finite–width neural networks trained with SGD and random augmentation.

We succeeded in our goal with Theorem 4.2, where we proved that, modulo an oft–satisfied $G$-invariance condition on the network architecture, ensembles become $G$-equivariant when training with SGD and random augmentation, assuming that the parameters of the network were distributed $G$-invariantly to begin with (e.g., with Gaussian initialization). That is, we have shown that if the goal is group equivariant model, it suffices to train a neural network ensemble model under data augmentation. Our numerical experiments, while not conclusive, support these conclusions for sample mean ensembles as well.

**Limitations**   Theorem 4.2 gives a guarantee of $G$-equivariance for 'infinitely large' ensembles. In practice the expected value in Equation 3 would need to be estimated with a finite ensemble mean. Furthermore, the $G$-invariance condition under which the result holds is only a sufficient condition. The same is true of the assumption that the initial distribution of parameters is $G$-invariant.

**Future Work**   Interesting directions for future work include bounding the error when using sample mean ensembles, as well as investigating the necessity of the $G$-invariance condition. In addition to this, it is an interesting question whether or not the initial distribution of parameters needs to be $G$-invariant or if the ensemble will still be 'pulled into' a $G$-equivariant state by the augmentation process.

**Reproducability statement**   We provide all the code used for the experiments in the supplementary material, along with detailed instructions to reproduce the experiments made in the paper.

**Ethics statement**   This work is of theoretical nature, and in particular do not involve human subjects, and only publicly available datasets. We see no obvious societal impact, positive or negative, in the short to middle term. The aim of the article is to provide new answers to the question when equivariance of a neural network model can be guaranteed only from training it on augmented data. This has relevance for the ongoing effort of providing more reliable and explainable AI. The authors declare that they have no conflicts of interest, and that they have followed good research practice.

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

## A  ASSORTED PROOFS

In this appendix we collect the proofs that we left out in the main test. Let us begin with the formula $\Phi_A(\rho_0(g)x) = \rho_N(g)\Phi_{\rho(g^{-1})A}(x)$. Note that this proof is merely a reformulation of the one given in Nordenfors et al. (2024) (see Lemma 3.9 of said article), and presented here only for convenience of the reader.

**Theorem A.1.** *Under the Assumption 1 that all non-linearities are equivariant, we have*

$$\Phi_A(\rho_0(g)x) = \rho_N(g)\Phi_{\rho(g^{-1})A}(x).$$

*Proof.* The proof is by induction on the number of layers $N$. Let $\Phi_A^N$ be the network given by the first $N$ layers of $\Phi_A$. Let us start with the case $N = 1$. For $N = 1$ we have

$$\rho_1(g)\Phi_{\rho(g^{-1})A}^1(x) = \rho_1(g)\sigma_0(\rho_1(g^{-1})A_0\rho_0(g^{-1})^{-1}x) = \sigma_0(\rho_1(g)\rho_1(g^{-1})A_0\rho_0(g^{-1})^{-1}x)$$

$$= \sigma_0(\mathrm{id}\, A_0\rho_0(g)x) = \Phi_A^1(\rho_0(g)x), \quad \forall g \in G,$$

where the first equality is by definition of $\rho(g)$, the second equality is by $G$-equivariance of $\sigma_0$, and the third equality follows since $\rho_i(g)$ is a representation, $\forall i \in \{0, 1\}, \forall g \in G$.

Now, assume that $\Phi_A^K(\rho_0(g)x) = \rho_K(g)\Phi_{\rho(g^{-1})A}^K(x)$, $\forall g \in G$. We want to show that $\Phi_A^{K+1}(\rho_0(g)x) = \rho_{K+1}(g)\Phi_{\rho(g^{-1})A}^{K+1}(x)$, $\forall g \in G$. We have

$$\rho_{K+1}(g)\Phi_{\rho(g^{-1})A}^{K+1}(x) = \rho_{K+1}(g)\sigma_K(\widehat{\rho}_K(g^{-1})A_K\Phi_{\rho(g^{-1})A}^K(x))$$

$$= \rho_{K+1}(g)\sigma_K(\rho_{K+1}(g^{-1})A_K\rho_K(g^{-1})^{-1}\Phi_{\rho(g^{-1})A}^K(x))$$

$$= \sigma_K(\rho_{K+1}(g)\rho_{K+1}(g^{-1})A_K\rho_K(g^{-1})^{-1}\Phi_{\rho(g^{-1})A}^K(x))$$

$$= \sigma_K(\mathrm{id}\, A_K\rho_K(g)\Phi_{\rho(g^{-1})A}^K(x))$$

$$= \sigma_K(A_K\Phi_A^K(\rho_0(g)x)) = \Phi_A^{K+1}(\rho_0(g)x), \quad \forall g \in G,$$

where the first equality is by the definition of $\Phi_A$, the second equality is by the definition of $\widehat{\rho}_K(g)$, the third equality follows by $G$-equivariance of $\sigma_K$, the fourth equality follows since $\rho_i(g)$ is a representation, $\forall i \in \{K, K+1\}$, $\forall g \in G$, the fifth equality follows by the inductive assumption, and the final equality is just the definition of $\Phi^{K+1}$.

By induction $\Phi_A(\rho_0(g)x) = \rho_N(g)\Phi_{\rho(g^{-1})A}(x)$, $\forall g \in G$. $\qquad\qquad\qquad\qquad\qquad$ □

We move on to the fact that $G$-invariant vector fields yield equivariant flows (Lemma 4.5), where we again give a proof only out of convenience for the reader – it is a slight reformulation of the proof of Theorem 2 in Köhler et al. (2020), but the statement should probably be well known to the experts.

**Theorem A.2.** *Let $V : \mathbb{R}^N \to \mathbb{R}^N$ be a $G$-equivariant, Lipschitz continuous vector field. Then, the flow $\mathcal{F}_t$ generated by $V$ also is.*

*Proof.* Let $x_0 \in \mathbb{R}^N$ be arbitrary. By definition of the flow, $\mathcal{F}_t(\rho(g)x_0)$ is the value of the unique solution $x^g$ to the initial value problem

$$\dot{x}^g(t) = V(x^g(t)), \quad x^g(0) = \rho(g)x_0. \tag{4}$$

Now let $x$ be the solution of

$$\dot{x}(t) = V(x), \quad x(0) = x_0.$$

Then, the function $u(t) = \rho(g)x(t)$ solves equation 4, since $u(0) = \rho(g)x(0) = \rho(g)x_0$, and

$$\dot{u}(t) = \rho(g)\dot{x}(t) = \rho(g)V(x(t)) = V(\rho(g)x(t)) = V(u(t)),$$

where we used the equivariance of $V$ in the penultimate step. Because of the uniqueness of the solution, we must then have $u(t) = x^g(t)$, i.e.

$$\mathcal{F}_t(\rho(g)x_0) = u(t) = \rho(g)x(t) = \rho(g)\mathcal{F}_t(x_0),$$

which is what was to be shown. $\qquad\qquad\qquad\qquad\qquad\qquad\qquad\qquad\qquad\qquad\qquad$ □

We move on to the proof of Lemma 4.1 i.e. that $\mathcal{L}$ is $G$-invariant if and only if $\Pi_{\mathcal{L}}$ is equivariant.

*Proof of Lemma 4.1.* Let us begin by proving the *only if* part. Assume that $\rho(g)\mathcal{L} \subseteq \mathcal{L}$, $\forall g \in G$. Clearly, if $\exists B \in \mathrm{T}\mathcal{L}$, $\exists g \in G$, such that $\rho(g)B \notin \mathrm{T}\mathcal{L}$, then $A = A_{\mathcal{L}} + B \in \mathcal{L}$, but $\rho(g)A = \rho(g)A_{\mathcal{L}} + \rho(g)B = A_{\mathcal{L}} + \rho(g)B \notin \mathcal{L}$. By contraposition, we must have $\rho(g)\mathrm{T}\mathcal{L} \subseteq \mathrm{T}\mathcal{L}$, $\forall g \in G$, since we have assumed that $\rho(g)\mathcal{L} \subseteq \mathcal{L}$, $\forall g \in G$. Now, suppose that $x \in \mathrm{T}\mathcal{L}$ and $y \in \mathrm{T}\mathcal{L}^{\perp}$. Then we have

$$\langle \rho(g)y, x \rangle = \langle y, \rho(g)^* x \rangle = \langle y, \rho(g)^{-1}x \rangle = \langle y, \rho(g^{-1})x \rangle = 0, \quad \forall g \in G,$$

where the second equality is by unitarity, the third equality is by definition of representations, and the final equality follows from $\rho(g)\mathrm{T}\mathcal{L} \subseteq \mathrm{T}\mathcal{L}$, $\forall g \in G$. Thus, it holds that $\rho(g)\mathrm{T}\mathcal{L}^{\perp} \subseteq \mathrm{T}\mathcal{L}^{\perp}$, $\forall g \in G$. We can decompose any $z \in \mathcal{H}$ as $z = z_{\mathrm{T}\mathcal{L}} + z_{\mathrm{T}\mathcal{L}^{\perp}}$, where $z_{\mathrm{T}\mathcal{L}} = \Pi_{\mathcal{L}}z$ and $z_{\mathrm{T}\mathcal{L}^{\perp}} = \Pi_{\mathcal{L}}^{\perp}z$. It follows that

$$\Pi_{\mathcal{L}}\rho(g)z = \Pi_{\mathcal{L}}\rho(g)(z_{\mathrm{T}\mathcal{L}} + z_{\mathrm{T}\mathcal{L}^{\perp}}) = \Pi_{\mathcal{L}}\rho(g)z_{\mathrm{T}\mathcal{L}} + \Pi_{\mathcal{L}}\rho(g)z_{\mathrm{T}\mathcal{L}^{\perp}} = \rho(g)z_{\mathrm{T}\mathcal{L}} = \rho(g)\Pi_{\mathcal{L}}z,$$

$\forall g \in G$, where the second equality is by linearity, the third equality follows by $\rho(g)\mathrm{T}\mathcal{L} \subseteq \mathrm{T}\mathcal{L}$ and $\rho(g)\mathrm{T}\mathcal{L}^{\perp} \subseteq \mathrm{T}\mathcal{L}^{\perp}$, $\forall g \in G$, and the last equality is by definition of $z_{\mathrm{T}\mathcal{L}}$. This concludes the *only if* part of the proof.

Let us now prove the *if* part of the proof. Assume that $\Pi_{\mathcal{L}}$ is $G$-equivariant. Then we have for any $B \in \mathrm{T}\mathcal{L}$ that

$$\rho(g)B = \rho(g)\Pi_{\mathcal{L}}B = \Pi_{\mathcal{L}}\rho(g)B \in \mathrm{T}\mathcal{L}, \quad \forall g \in G,$$

since $\Pi_{\mathcal{L}}$ is a $G$-equivariant projection onto $\mathrm{T}\mathcal{L}$. Thus, we have for any $A \in \mathcal{L}$ that

$$\rho(g)A = \rho(g)(A_{\mathcal{L}} + B) = \rho(g)A_{\mathcal{L}} + \rho(g)B = A_{\mathcal{L}} + \rho(g)B \in \mathcal{L}, \quad \forall g \in G,$$

since $\mathcal{L} = A_{\mathcal{L}} + \mathrm{T}\mathcal{L}$ with $\rho(g)A_{\mathcal{L}} = A_{\mathcal{L}}$ and $\rho(g)$ is linear. $\qquad\qquad\qquad\qquad\qquad$ □

## A.1 $G$-INVARIANT INITIALIZATION OF THE NETWORK

Let us now, as advertised in the main paper, prove that a Gaussian initialization of the coefficients $c$ will, as soon as $\mathcal{L}$ is $G$-invariant, yield a $G$-invariant distributions of the parameters $A$,

**Lemma A.3.** *Let $L : \mathbb{R}^p \to \mathcal{L}$ be a parametrization of $\mathcal{L}$, $\mathcal{L} = A_\mathcal{L} + L\mathbb{R}^p$, with $A_\mathcal{L} \in \mathcal{H}_G$. Then, if $c$ is a standard Gaussian vector (with i.i.d. entries), and $\mathcal{L}$ is $G$-invariant, $A = A_\mathcal{L} + Lc$ will have a $G$-invariant distribution.*

*Proof.* Because $c$ is standard Gaussian, $A$ will also have a Gaussian distribution, with mean $A_\mathcal{L}$ and covariance matrix $LL^* = \Pi_\mathcal{L}$, where the latter follows from the fact that $L$ is unitary. Now, by the same argument, $\rho(g)A = \rho(g)A_\mathcal{L} + \rho(g)Lc$ will be Gaussian with mean $\rho(g)A_\mathcal{L} = A_\mathcal{L}$ – remember that $A_\mathcal{L} \in \mathcal{H}_G$ – and covariance $\rho(g)L(\rho(g)L)^*$. We may now argue

$$\rho(g)L(\rho(g)L)^* = \rho(g)LL^*\rho(g)^* = \rho(g)\Pi_\mathcal{L}\rho(g)^* = \Pi_\mathcal{L}\rho(g)\rho(g)^* = \Pi_\mathcal{L}.$$

The penultimate step follows from the $G$-invariance of $\mathcal{L}$, and the final one from the unitarity of $\rho(g)$. This shows that $A$ and $\rho(g)A$ both are Gaussians with mean $A_\mathcal{L}$ and covariance $\Pi_\mathcal{L}$, and we are done. $\square$

## A.2 A RESULT ABOUT THE GRADIENTS OF RANDOM FUNCTIONS

In preparation of our proof of the stochastical version of the main result, let us prove the following very intuitive lemma about random functions: If two random functions, that almost surely are differentiable, have the property that their pointwise distributions are equal, the same is true for their gradients. For the proof, we will invoke the Cramér-Wold Theorem (Cramér & Wold, 1936).

**Theorem A.4** (Cramér-Wold). *A probability distribution $p$ on $\mathbb{R}^d$ is uniquely determined by the set of distributions $p \circ \pi_v^{-1}$ on $\mathbb{R}$, indexed by the unit vector $v$, where $\pi_v : \mathbb{R}^d \to \mathbb{R}$, $x \mapsto \langle x, v \rangle$.*

We now formulate and prove the statement.

**Lemma A.5.** *Let $f$ and $g$ be random functions $\mathbb{R}^d \to \mathbb{R}$ that almost surely are $C^1$. If for every $x \in \mathbb{R}^d$, $f(x) \sim g(x)$, then also $\nabla f(x) \sim \nabla g(x)$, for all $x \in \mathbb{R}^d$.*

*Proof.* Assume that $f(x) \sim g(x)$, for all $x \in \mathbb{R}^d$. Then

$$X_n := \frac{f(x + \frac{1}{n}v) - f(x)}{\frac{1}{n}} \sim \frac{g(x + \frac{1}{n}v) - g(x)}{\frac{1}{n}} =: Y_n, \quad \forall x, v \in \mathbb{R}^d, n \in \mathbb{Z}_+.$$

Now, for every $x \in \mathbb{R}^d$ and every $v \in S^{d-1}$, $X_n \to \langle \nabla f(x), v \rangle$ and $Y_n \to \langle \nabla_x g(x), v \rangle$ a.s. as $n \to \infty$. Thus, the distributions of the limit random variables (i.e., the distributions of the directional derivative in direction $v$) also agree. So that

$$\langle \nabla f(x), v \rangle \sim \langle \nabla g(x), v \rangle, \quad \forall x \in \mathbb{R}^d, \forall v \in S^{d-1}.$$

Since this holds for every $x \in \mathbb{R}^d$ and every $v \in S^{d-1}$, we can apply the Cramér-Wold Theorem to conclude that

$$\nabla f(x) \sim \nabla g(x), \quad \forall x \in \mathbb{R}^d.$$

$\square$

# B PROOF OF THEOREM 4.2 IN THE CASE OF SGD WITH RANDOM AUGMENTATION

In this appendix we will prove the main result of this paper in the stochastic setting. We will proceed by proving a few lemmas, and finally proving the main result.

In Section 4 we utilize the fact that a $G$-invariant function has a $G$-equivariant gradient. Now, $R^g$ is clearly not $G$-invariant for any fixed draw of the $g_k$ and $x_k$, but as we will see, it is however in a sense $G$-invariant *in distribution*. We will now prove that the implication holds for equality in distribution for random functions.

**Lemma B.1.** *If $f$ is a random function $\mathcal{H} \to \mathbb{R}$ that almost surely is $C^1$ and satisfies $f(A) \sim (f \circ \rho(g))(A)$, for every $g \in G$ and every $A \in \mathcal{H}$, then $\nabla f$ satisfies $\nabla f(\rho(g)A) \sim \rho(g)\nabla f(A)$, for every $g \in G$ and every $A \in \mathcal{H}$.*

*Proof.* We have that

$$
\begin{aligned}
\nabla f(\rho(h)A) &\sim \nabla(f \circ \rho(g))(\rho(h)A) \\
&= \rho(g)^* \nabla f(\rho(g)\rho(h)A) \\
&= \rho(h)\rho(h)^*\rho(g)^* \nabla f(\rho(g)\rho(h)A) \\
&= \rho(h)\rho(gh)^* \nabla f(\rho(gh)A) \\
&= \rho(h)\nabla(f \circ \rho(gh))(A) \sim \rho(h)\nabla f(A), \quad \forall h \in G,
\end{aligned}
$$

where the first equality in distribution follows from Lemma A.5 with $f = f$ and $g = f \circ \rho(g)$, the second is simply the chain rule, the third follows because $\rho(h)$ is assumed unitary so that $\rho(h)\rho(h)^* = \mathrm{id}$, the fourth because $\rho$ is a representation, the fifth is again simply the chain rule, and since $gh \in G$, the final equality in distribution follows by Lemma A.5 with $f = f$ and $g = f \circ \rho(gh)$. $\qquad\square$

We will now show that $R^g$ satisfies $R^g(A) \sim (R^g \circ \rho(g))(A)$, for every $g \in G$ and every $A \in \mathcal{H}$, and thus that the gradient of $R^g$ satisfies $\nabla R^g(\rho(g)A) \sim \rho(g)\nabla R^g(A)$, for every $g \in G$ and every $A \in \mathcal{H}$. This is the counterpart of Lemma 4.6 for SGD with random augmentation.

**Lemma B.2.** *The gradient of $R^g$ satisfies $\nabla R^g(\rho(g)A) \sim \rho(g)\nabla R^g(A)$, for every $g \in G$ and every $A \in \mathcal{H}$.*

*Proof.* We want to show that $\nabla R^g(\rho(g)A) \sim \rho(g)\nabla R^g(A)$, for every $g \in G$ and every $A \in \mathcal{H}$. By Lemma B.1 it suffices to show that $R^g(A) \sim (R^g \circ \rho(g))(A)$, for every $g \in G$ and every $A \in \mathcal{H}$. Applying Theorem A.1, we get that for every $A \in \mathcal{H}$ that

$$
\begin{aligned}
R^g(\rho(h)A) &= \frac{1}{s}\sum_{k=1}^{s} \ell(\Phi_{\rho(h)A}(\rho_0(g_k)x_k), \rho_N(g_k)y_k) \\
&= \frac{1}{s}\sum_{k=1}^{s} \ell(\rho_N(h)\Phi_A(\rho_0(h)^{-1}\rho_0(g_k)x_k), \rho_N(g_k)y_k)
\end{aligned}
$$

Invoking the $G$-invariance of $\ell$ and then that $\rho$ is a representation, the above can be rewritten to

$$
\frac{1}{s}\sum_{k=1}^{s} \ell(\Phi_A(\rho_0(h)^{-1}\rho_0(g_k)x_k), \rho_N(h)^{-1}\rho_N(g_k)y_k)
$$

Now, by the invariance property of the Haar measure, the tuple of group elements $g_0, \ldots g_{s-1}$ are equidistributed with the tuple $h^{-1}g_0, \ldots h^{-1}g_{s-1}$. Since the data points $(x_k, y_k)$ are inpendendent of the draw of the $g_k$, we can conclude that the above is equidistributed with

$$
\frac{1}{s}\sum_{k=1}^{s} \ell(\Phi_A(\rho_0(g_k)x_k), \rho_N(g_k)y_k) = R^g(A),
$$

which is what we wanted to show. $\qquad\square$

We need only prove one additional lemma before we can prove our main result. Namely, we need to show that if we begin with $G$-invariantly distributed parameters $A^0$, then SGD with random augmentation will always lead to new parameters $A^t$ which are $G$-invariantly distributed. This is the counterpart to Lemma 4.4 for the case of SGD with random augmentation.

**Lemma B.3.** *If $\rho(g)A^0 \sim A^0$, for every $g \in G$, and $\mathcal{L}$ is $G$-invariant, then $\rho(g)A^t \sim A^t$, for every $g \in G$, for all $t$.*

*Proof.* We proceed by induction. Assume that $\rho(g)A^0 \sim A^0$ for every $g \in G$ and $\mathcal{L}$ is $G$-invariant. The base case thus follows by assumption, i.e., $\rho(g)A^0 \sim A$, for every $g \in G$. Let us now carry out the inductive step. We have that if $\rho(g)A^t \sim A^t$, for every $g \in G$, then

$$
\begin{aligned}
\rho(h)A^{t+1} &= \rho(h)(A^t - \gamma_t \Pi_{\mathcal{L}} \nabla R^g(A^t)) \\
&= \rho(h)A^t - \gamma_t \rho(h)\Pi_{\mathcal{L}} \nabla R^g(A^t) \\
&= \rho(h)A^t - \gamma_t \Pi_{\mathcal{L}} \rho(h) \nabla R^g(A^t) \\
&\sim \rho(h)A^t - \gamma_t \Pi_{\mathcal{L}} \nabla R^g(\rho(h)A^t) \\
&\sim A^t - \gamma_t \Pi_{\mathcal{L}} \nabla R^g(A^t)) = A^{t+1}, \quad \forall h \in G,
\end{aligned}
$$

where the first equality is by the recursion formula for SGD with random augmentation, the second equality is by linearity of $\rho(h)$, the third equality follows from Lemma 4.1 since we have assumed that $\mathcal{L}$ is $G$-invariant, the fourth equality in distribution follows from Lemma B.2, the fifth equality in distribution is by our inductive assumption, and the final equality is again by the recursion formula for SGD with random augmentation. Thus, it follows by induction that $\rho(g)A^t \sim A^t$, $\forall g \in G$, $\forall t$. $\qquad \square$

We may now prove the stochastic version of our main result.

*Proof of Theorem 4.2 in the case of SGD with random augmentation.* Assume that $\rho(g)A^0 \sim A^0$ and $\rho(g)\mathcal{L} \subseteq \mathcal{L}$, $\forall g \in G$. Thus, by Lemma B.3 we have that $\rho(g)A^t \sim A^t$, $\forall g \in G$, by which it follows from Lemma 4.3 that $\overline{\Phi}_t$ is $G$-equivariant. $\qquad \square$

## C  CONVOLUTIONS WITH ASYMMETRIC FILTERS AND $C_4$-INVARIANCE

In the main paper, we have used $\mathcal{L}^{\mathrm{sym}}$ and $\mathcal{L}^{\mathrm{as}}$ as running examples. We have argued that if the canonical representation $\rho^{\mathrm{rot}}$ is used on all intermediate spaces, $\mathcal{L}^{\mathrm{sym}}$ is invariant under all transformations $\rho(g)$, whereas $\mathcal{L}^{\mathrm{as}}$ is not. However, as we remarked in Example 4.1, there is no inherent reason why the actions on the intermediate spaces should always be $\rho^{\mathrm{rot}}$. The purpose of this section is to show that if we fix the representation $\rho^{\mathrm{rot}}$ on $X$, it is not. To do this, it is clearly enough to focus on the space of convolutional maps between $X_0 = \mathbb{R}^{n,n}$ and $X_1 = (\mathbb{R}^{n,n})^d$.

To make the case clear that this is an important and nontrivial point, let us first show that there are representations other than $\rho^{\mathrm{rot}}$ on $X_1$ which makes $\mathcal{L}^{\mathrm{sym}}$ invariant. Let $\rho^{\mathrm{ch}}$ be a representation of $C_4$ on $\mathbb{R}^d$. Such exist: Note that $C_4$ is isomorphic to $\mathbb{Z}_4$, the integers equipped with addition modulo 4, and $\mathbb{Z}_4$ naturally acts on the space $\mathbb{R}^4$ by shifting the entries, i.e $(\varrho(k)v)_\ell = v_{\ell-k}$, which then also does $C_4$. In obvious ways, this can be extended to any $\mathbb{R}^d$. For any representation $\rho^{\mathrm{ch}}$ of $C_4$ on $\mathbb{R}^d$, we can combine it with $\rho^{\mathrm{rot}}$ as follows:

$$
\left((\rho^{\mathrm{ch}} \odot \rho^{\mathrm{rot}})(g)x\right)_k = \left( \sum_{\ell \in [d]} \rho^{\mathrm{ch}}(g)_{k\ell} \rho^{\mathrm{rot}}(g)x_\ell \right)_k. \tag{5}
$$

In words, $\rho^{\mathrm{ch}} \odot \rho^{\mathrm{rot}}$ first rotates each entry of a tuple $x$ according to $\rho^{\mathrm{ch}}(g)$, and then transforms the $d$ resulting $\mathbb{R}^{N,N}$-images by $\rho^{\mathrm{ch}}$ as if they were entries in a $\mathbb{R}^d$-vector. A straightforward calculation now shows that the lifted representation $\widehat{\rho_0}(g)$ then maps $\langle \psi \rangle$ to

$$
\widehat{\rho_0}(g)\langle \psi \rangle = \langle \left( \sum_\ell \rho^{\mathrm{ch}}(g)_{k\ell} \rho^{\mathrm{rot}}(g)\psi_\ell \right)_k \rangle. \tag{6}
$$

Consequently, if the filters $\psi_\ell$ are symmetric filters, the transformed ones are also. On the other hand, if the $\psi$ are supported on the asymmetric support, the linear combinations in equation 6 will in general not be, and hence $\rho(g)\mathcal{L} \not\subseteq \mathcal{L}$ also for intermediate representations of the form equation 5.

The last point opens up a route to prove that there are no representations $\rho_1$ at all that makes $\mathcal{L}^{\mathrm{as}}$ invariant under the corresponding lifted representation: If we show that if $\rho_1$ is a representation for which $\rho(g)\mathcal{L}^{\mathrm{as}} \subseteq \mathcal{L}^{\mathrm{as}}$ for all $g$, then it must be as in equation 5, then we are by the previous discussion done. This is the purpose of the following theorem.

**Theorem C.1.** *Let $N \geq 3$, and let $C_4$ act through $\rho^{\mathrm{rot}}$ on $X$. If $\rho_1$ is a representation such that $\mathcal{L}^{\mathrm{as}}$ is $G$-invariant under the lifted representation, it must be as in equation 5. Since no such representations make $\mathcal{L}^{\mathrm{as}}$ $G$-invariant, there are no representations of $C_4$ on $X_1$ that make $\mathcal{L}^{\mathrm{as}}$ $G$-invariant.*

*Proof.* For any representation $\rho_1$ on $X_1$, we have

$$\rho(g)\langle\varphi\rangle = \rho_1(g)\langle\varphi\rangle\rho^{\mathrm{rot}}(g)^{-1} = \rho_1(g)\rho^{\mathrm{rot}}(g)^{-1}\rho^{\mathrm{rot}}(g)\langle\varphi\rangle\rho^{\mathrm{rot}}(g)^{-1}$$
$$= \rho_1(g)\rho^{\mathrm{rot}}(g)^{-1}\langle\rho^{\mathrm{rot}}(g)\varphi\rangle.$$

Now, we assume that $\rho_1(g)$ is a representation for which $\rho(g)\mathcal{L} \subseteq \mathcal{L}$. Then, the above operator must be of the form $\langle\psi\rangle$ for all $\varphi$ and $g$. Since convolutions commute with any translation $T_\ell$, we can then argue that

$$T_\ell\rho_1(g)\rho^{\mathrm{rot}}(g)^{-1}\langle\rho^{\mathrm{rot}}(g)\varphi\rangle = T_\ell\langle\psi\rangle = \langle\psi\rangle T_\ell = \rho_1(g)\rho^{\mathrm{rot}}(g)^{-1}\langle\rho^{\mathrm{rot}}(g)\varphi\rangle T_\ell$$
$$= \rho_1(g)\rho^{\mathrm{rot}}(g)^{-1}T_\ell\langle\rho^{\mathrm{rot}}(g)\varphi\rangle \quad (7)$$

where we again did not distinguish between translation representation on $X$ and the direct product of them on $X_1$. Since this equality is true for any $\varphi$, this implies that

$$T_\ell\rho_1(g)\rho^{\mathrm{rot}}(g)^{-1} = \rho_1(g)\rho^{\mathrm{rot}}(g)^{-1}T_\ell \quad (8)$$

for all $g$ and $\ell$. A more technical argument goes as follows: By choosing $\varphi$ equal to the tuple with only a non-trivial filter in the $i$:th channel, that filter to have only one non-zero pixel, and subsequently evaluating equation 7 on a basis of $X$, we get the desired equality of operators on a basis of $X_1$.

Now, equation 8 simply means that the operator $\rho_1(g)\rho^{\mathrm{rot}}(g)^{-1}$ commutes with translations for every $g$. It is well known that this is equivalent to $\rho_1(g)\rho^{\mathrm{rot}}(g)^{-1}$ being a convolution operator $\langle\chi(g)\rangle$ for some filters $\chi_{k\ell}(g)$, $k,\ell \in [d]$. Inserting this form into the definition of $\rho$ yields

$$\rho(g)\langle\varphi\rangle = \langle\chi\rangle\langle\rho^{\mathrm{rot}}(g)\varphi\rangle = \left\langle \sum_{\ell\in[d]} \chi_{k\ell}(g) * (\rho^{\mathrm{rot}}(g)\varphi_\ell) \right\rangle \quad (9)$$

Now, suppose that any filter $\chi_{k\ell}(g)$ has a support of more than one pixel. Then, since $N \geq 3$, we can construct a filter $\varphi_\ell$ supported on the asymmetric set so that the support of $\chi_{k\ell}(g) * (\rho^{\mathrm{rot}}(g)\varphi_\ell)$ is not – essentially, the convolution by $\chi_{k\ell}(g)$ would spread out the support of $\varphi_\ell$. By letting all other filters in a tuple be zero, we would conclude that the filter

$$\sum_{\ell\in[d]} \chi_{k\ell}(g) * (\rho^{\mathrm{rot}}(g)\varphi_\ell)$$

has a support which is not contained in the asymmetric $\Omega_0$. Together with equation 9, this means that $\rho(g)\langle\varphi\rangle \notin \mathcal{L}^{\mathrm{as}}$, which would be a contradiction. Hence, all $\langle\chi_g\rangle$ must be multiples of the identity for all $g$, which means that

$$[\rho_1(g)v]_k = \sum_{\ell\in[d]} c_{k\ell}(g)\rho^{\mathrm{rot}}(g)x_\ell$$

for some numbers $c_{k\ell}(g)$. It is now only left to show that the $c_{k\ell}$ define a representation. We however have for $g, h \in C_4$ arbitrary

$$[\rho_1(gh)x]_k = \sum_{\ell\in[d]} c_{k\ell}(gh)\rho^{\mathrm{rot}}(gh)x_\ell$$

$$[\rho_1(g)\rho_1(h)x]_k = \sum_{\ell\in[d]} c_{k\ell}(g)\rho^{\mathrm{rot}}(g)(\rho_1(g)x)_\ell = \sum_{\ell,m\in[d]} c_{km}(g)\rho^{\mathrm{rot}}(g)c_{m\ell}(h)\rho^{\mathrm{rot}}(h)x_\ell$$

Since $\rho_1$ and $\rho^{\mathrm{rot}}$ are representations, the above expressions are equal for every $x$ and $g$, which means that

$$\forall k,\ell \in [d]: \quad c_{k\ell}(gh) = \sum_{m\in[d]} c_{km}(g)c_{m\ell}(h)$$

This is however only another way of saying that the matrices $C(g) = (c_{k\ell}(g))_{k,\ell}$ fulfill $C(gh) = C(g)C(h)$, which is what was to be shown. $\square$

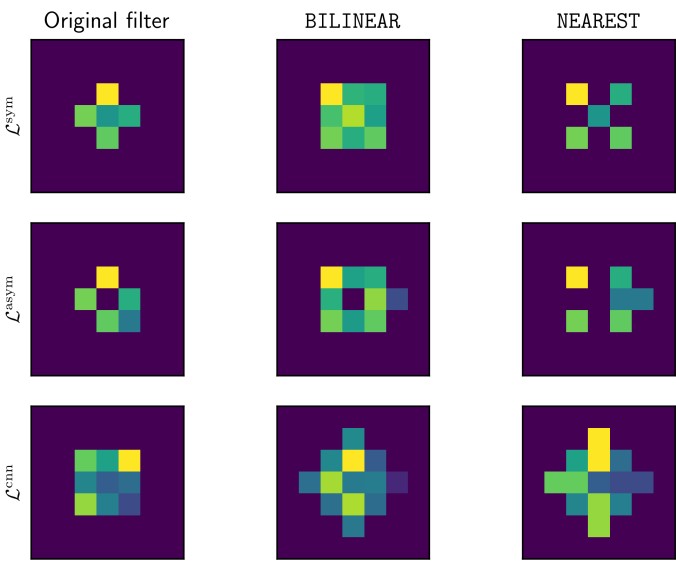

Figure 6: The effects of different types of interpolation on different types of filters. Filters in $\mathcal{L}^{\mathrm{sym}}$, $\mathcal{L}^{\mathrm{as}}$ and $\mathcal{L}^{\mathrm{cnn}}$ are rotated 45 degrees using the `BILINEAR` and `NEAREST` interpolations.

## D    EXPERIMENT DETAILS

### D.1    HARDWARE

The $C_4$-experiments were performed on a cluster using NVIDIA Tesla T4 GPUs with 16GB RAM per unit. The total compute time for training all 3000 individual models for 10 epochs each is estimated at $\sim 115$ hours. The $C_{16}$-experiments were made on the same cluster, but instead using NVIDIA Tesla A40 GPUs with 64GB RAM per unit. These experiments (`BILINEAR` and `NEAREST`) used a total of $\sim 700$ compute hours.

### D.2    ARCHITECTURE

The architecture used for all three networks is the same with the exception of the support of the $3 \times 3$ filters. Namely, in the first layer we use a convolution with zero padding, with 1 channel in and 16 out, followed by an average pooling with a $2 \times 2$ window with a stride of 2, followed by a `tanh` activation, followed by a layer normalization. The second layer is the same except the convolution has 16 channels in instead of 1. The third layer is the same as the second, but without the average pooling. The final layer flattens the image channels into a vector, followed by a linear transformation into $\mathbb{R}^{10}$.

All of the nonlinearities involved here are equivariant to $C_4$- rotations, and so do not interfere with the equivariance/invariance of the network in that case. Due to interpolation effects, it is not equivariant to $C_{16}$ – adding to the assumptions of our theorem that are violated in that case.

A prediction of the network is then the `argmax` of the output, which yields the predicted label as a one-hot vector.

## E    $C_{16}$ WITH `NEAREST` INTERPOLATION

We repeat the experiment in the main paper while using `NEAREST` interpolation scheme. The results are given in Figure 7. Again, none of the models fare particularly well, as the theory suggests.

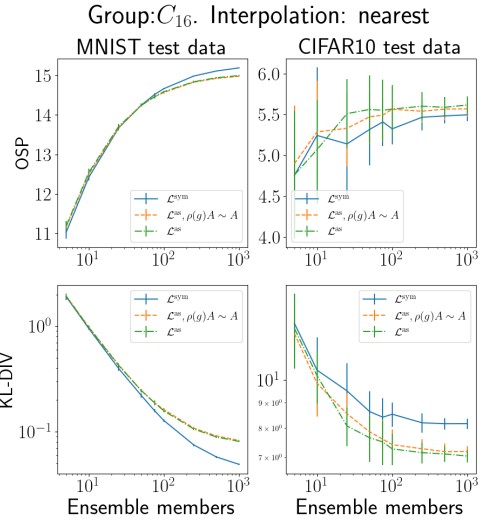

Figure 7: Metrics after the 10th epochs for different ensemble sizes for the $C_{16}$ experiment using the `NEAREST` interpolation.
Each datapoint is a mean of 30 bootstrapped examples – the error-bars denotes one standard deviation of the bootstrap.
The $x$-scale in the top plots are logaritmic, both scales are logaritmic in the bottom plots. Best viewed in color.

Compared to the `BILINEAR` interpolation experiment, see Figure 5 (right), we see that the models fare better on the MNIST data, but worse on the CIFAR data. We think that the reason for the better performance of the indistribution data is that the `NEAREST` interpolation introduces fewer 'bluriness' artefacts than the `BILINEAR` one – the former still operates by permuting pixels. This means that the test set formed by applying `BILINEAR`-rotations to MNIST results in a more diverse dataset – which makes it simpler to fit to it.

We speculate that the reason for the worse performance on the CIFAR data is that the the operators $\Pi_{\mathcal{L}}$ and $\rho_i(g)$ are even further from commuting in the $C_{16}$ case compared to the $C_4$ case – see Figure 6 .

## F EXPERIMENT WITH LARGER FILTERS

We perform the same experiment as in Section 5 with convolution filters of size $5 \times 5$. The reason for performing this experiment is to test whether the 'small' error in the approximation $\rho(g)\Pi_{\mathcal{L}}A \approx \Pi_{\mathcal{L}}\rho(g)A$ in the case of $3 \times 3$-filters is responsible for the small difference in performance between symmetric and asymmetric models. In a larger filter we can have more energy in the asymmetric parts of the filter, so we want to see if this affects the results. We consider therefore two CNNs with filters as in Figure 8.

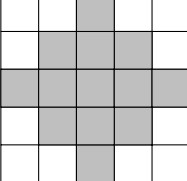 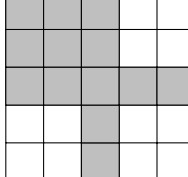

Figure 8: Left: symmetric filter. Right: asymmetric filter. Grey indices correspond to indices where the filter is supported.

As before, we train under augmentation by multiples $\pi/2$ radian rotations of our input images, and to keep the dimensions of input and output the same, we have to modify the convolution to have 2 rows of zero padding. The asymmetric filters are initialized with a $G$-invariant distribution by setting the four indices in the top left corner to 0, in addition to initializing with a standard normal distribution.

The results from this experiment can be seen in Figure 9 and Figure 10, where we show the results of the experiment in Section 5 and the experiment in this section side-by-side. We also give the metrics after epoch 10 in Table 2

We can see that the difference in performance, with regards to the metrics of OSP and divergence, between the symmetric and asymmetric models is larger in the case of the $5 \times 5$–filters, which is what one would predict if the reason for the size of the discrepancy between symmetric and asymmetric models is the size of the error when approximating $\rho(g)\Pi_{\mathcal{L}} A \approx \Pi_{\mathcal{L}}\rho(g)A$.

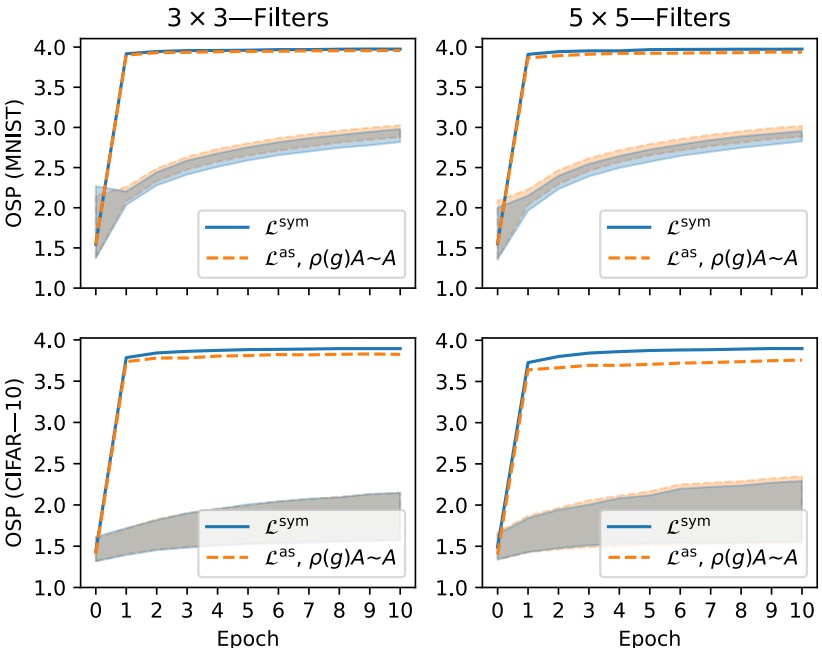

Figure 9: Top: OSP on MNIST test data for ensembles with 1000 members (higher is better). The middle $95\%$ of individual ensemble members are within the shaded area.
Bottom: OSP on CIFAR–10 test data for ensembles with 1000 members (higher is better). The middle $95\%$ of individual ensemble members are within the shaded area. Best viewed in color.

| Model | MNIST | | CIFAR–10 | |
|---|---|---|---|---|
| | OSP | $\log D_{\text{KL}}$ | OSP | $\log D_{\text{KL}}$ |
| $\mathcal{L}^{\text{sym}}$ | **3.97** | **−2.35** | **3.90** | **−1.50** |
| $\mathcal{L}^{\text{as}}, \rho(g)A \sim A$ | 3.94 | −1.54 | 3.76 | −0.54 |

Table 2: Metrics after the $10^{\text{th}}$ epoch of training for ensembles with 1000 members using $5 \times 5$-filters. Standard deviations are over test data.

## G   DETAILED RESULTS FOR DIFFERENT ENSEMBLE SIZES

We here present our results in table form: the $C_4$-experiments in Table 3, $C_{16}$-experiments with BILINEAR interpolation in Table 4 and $C_{16}$-experiments with NEAREST interpolation in Table 5. We indicate models that perform statistically significantly better than all others (according to a $t$-test of the 30 bootstrapped examples) than all others of the experiments ($p < .001$) with bold font.

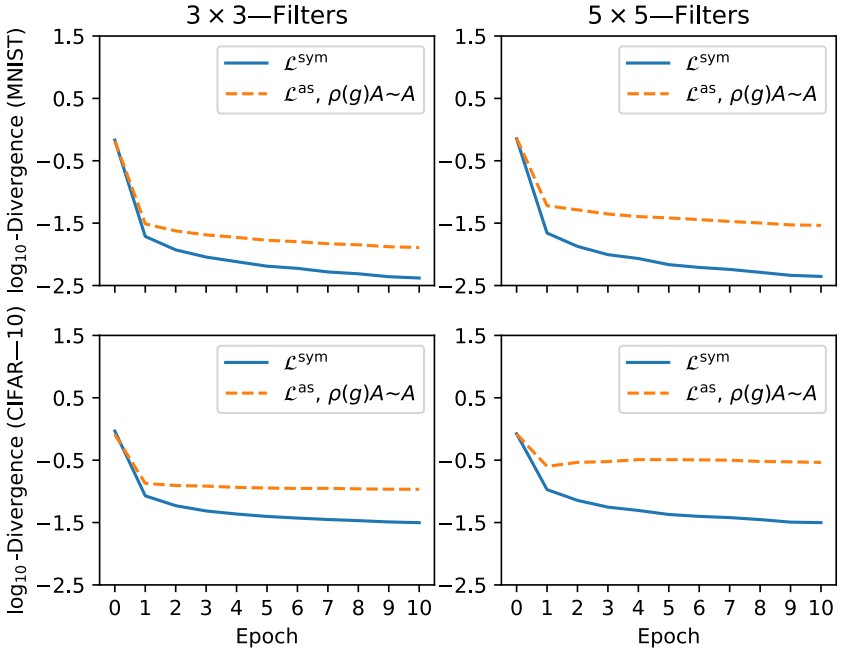

Figure 10: Top: Kullback–Leibler divergence on MNIST test data for ensembles with 1000 members (lower is better).
Bottom: Kullback–Leibler divergence on CIFAR–10 test data for ensembles with 1000 members (lower is better). Best viewed in color.

| | Metric | Model | 5 | 10 | 25 | 50 | 75 | 100 | 250 | 500 | 1000 |
|---|---|---|---|---|---|---|---|---|---|---|---|
| MNIST | OSP | $\mathcal{L}^{\text{sym}}$ | 3.587 | 3.722 | 3.830 | 3.879 | 3.902 | 3.914 | **3.944** | **3.957** | **3.964** |
| | | $\mathcal{L}^{\text{as}}$, (s.in) | **3.619** | **3.745** | **3.841** | **3.885** | 3.903 | 3.913 | 3.936 | 3.945 | 3.951 |
| | | $\mathcal{L}^{\text{as}}$ | 3.589 | 3.714 | 3.811 | 3.854 | 3.870 | 3.879 | 3.897 | 3.904 | 3.907 |
| | $\log D_{\text{KL}}$ | $\mathcal{L}^{\text{sym}}$ | -0.31 | -0.65 | -1.06 | -1.36 | -1.53 | -1.65 | -1.99 | -2.22 | -2.39 |
| | | $\mathcal{L}^{\text{as}}$, (s.in) | -0.33 | -0.67 | -1.06 | -1.34 | -1.49 | -1.58 | -1.85 | -1.98 | -2.08 |
| | | $\mathcal{L}^{\text{as}}$ | -0.29 | -0.59 | -0.94 | -1.15 | -1.26 | -1.32 | -1.45 | -1.51 | -1.54 |
| CIFAR-10 | OSP | $\mathcal{L}^{\text{sym}}$ | 2.656 | 2.967 | 3.361 | 3.536 | 3.620 | 3.661 | 3.775 | 3.823 | 3.854 |
| | | $\mathcal{L}^{\text{as}}$, (s.in) | 2.707 | 3.019 | 3.382 | 3.548 | 3.614 | 3.652 | 3.746 | 3.778 | 3.800 |
| | | $\mathcal{L}^{\text{as}}$ | 2.698 | 3.068 | 3.383 | 3.493 | 3.552 | 3.576 | 3.620 | 3.639 | 3.649 |
| | $\log D_{\text{KL}}$ | $\mathcal{L}^{\text{sym}}$ | 0.52 | 0.25 | -0.17 | -0.46 | -0.64 | -0.74 | -1.10 | -1.32 | -1.49 |
| | | $\mathcal{L}^{\text{as}}$, (s.in) | 0.52 | 0.24 | -0.17 | -0.44 | -0.58 | -0.68 | -0.96 | -1.08 | -1.16 |
| | | $\mathcal{L}^{\text{as}}$ | 0.54 | 0.26 | -0.09 | -0.27 | -0.37 | -0.42 | -0.51 | -0.56 | -0.58 |

Table 3: Results for the $C_4$ experiment. Shown are the metrics measured for different ensemble sizes at the last epoch of training. (s.in) refers to the symmetrical initialization. The numbers presented are means of 30 bootstrapped ensembles of the respective sizes. Bold results that are significantly better (as measured by a $t$-test, $p < .001$) than all other models at the respective metric and size.

| | Metric | Model | 5 | 10 | 25 | 50 | 75 | 100 | 250 | 500 | 1000 |
|---|---|---|---|---|---|---|---|---|---|---|---|
| MNIST | OSP | $\mathcal{L}^{\text{sym}}$ | 7.80 | 8.92 | 9.71 | 10.13 | 10.23 | 10.29 | 10.44 | 10.50 | 10.55 |
| | | $\mathcal{L}^{\text{as}}$, (s.in) | 8.64 | 9.64 | 10.51 | 11.01 | 11.11 | 11.26 | 11.35 | 11.44 | 11.47 |
| | | $\mathcal{L}^{\text{as}}$ | 8.54 | 9.63 | 10.54 | 10.91 | 11.09 | 11.15 | 11.30 | 11.44 | 11.43 |
| | | $\mathcal{L}^{\text{cnn}}$ | **10.79** | **12.25** | **13.52** | **14.11** | **14.34** | **14.47** | **14.75** | **14.87** | **14.93** |
| | $\log D_{\text{KL}}$ | $\mathcal{L}^{\text{sym}}$ | 0.83 | 0.69 | 0.58 | 0.52 | 0.51 | 0.51 | 0.49 | 0.49 | 0.48 |
| | | $\mathcal{L}^{\text{as}}$, (s.in) | 0.70 | 0.55 | 0.43 | 0.36 | 0.34 | 0.32 | 0.31 | 0.30 | 0.30 |
| | | $\mathcal{L}^{\text{as}}$ | 0.72 | 0.56 | 0.42 | 0.37 | 0.33 | 0.33 | 0.31 | 0.29 | 0.29 |
| | | $\mathcal{L}^{\text{cnn}}$ | **0.40** | **0.11** | **-0.25** | **-0.49** | **-0.59** | **-0.66** | **-0.82** | **-0.89** | **-0.93** |
| CIFAR | OSP | $\mathcal{L}^{\text{sym}}$ | 5.30 | 6.20 | 6.72 | 7.19 | 7.21 | 7.29 | 7.40 | 7.56 | 7.52 |
| | | $\mathcal{L}^{\text{as}}$, (s.in) | 6.00 | 6.95 | 7.51 | 7.60 | 7.91 | 8.01 | 7.96 | 8.02 | 8.08 |
| | | $\mathcal{L}^{\text{as}}$ | 5.79 | 6.73 | 7.60 | 7.88 | 7.88 | 8.02 | 8.07 | 8.12 | 8.18 |
| | | $\mathcal{L}^{\text{cnn}}$ | **8.09** | **9.88** | **11.46** | **12.49** | **12.68** | **12.80** | **13.17** | **13.20** | **13.28** |
| | $\log D_{\text{KL}}$ | $\mathcal{L}^{\text{sym}}$ | 1.04 | 0.91 | 0.81 | 0.76 | 0.76 | 0.74 | 0.73 | 0.72 | 0.72 |
| | | $\mathcal{L}^{\text{as}}$, (s.in) | 0.99 | 0.85 | 0.74 | 0.70 | 0.67 | 0.66 | 0.65 | 0.64 | 0.64 |
| | | $\mathcal{L}^{\text{as}}$ | 1.00 | 0.86 | 0.73 | 0.68 | 0.67 | 0.65 | 0.64 | 0.63 | 0.63 |
| | | $\mathcal{L}^{\text{cnn}}$ | **0.87** | **0.67** | **0.47** | **0.34** | **0.32** | **0.31** | **0.26** | **0.26** | **0.25** |

Table 4: Results for the $C_{16}$ experiment with `BILINEAR` interpolation. Shown are the metrics measured for different ensemble sizes at the last epoch of training. (s.in) refers to the symmetrical initialization. TThe numbers presented are means of 30 bootstrapped ensembles of the respective sizes. Bold results that are significantly better (as measured by a $t$-test, $p < .001$) than all other models at the respective metric and size

| | Metric | Model | 5 | 10 | 25 | 50 | 75 | 100 | 250 | 500 | 1000 |
|---|---|---|---|---|---|---|---|---|---|---|---|
| MNIST | OSP | $\mathcal{L}^{\text{sym}}$ | 11.02 | 12.42 | 13.67 | 14.25 | **14.52** | **14.67** | **14.99** | **15.11** | **15.19** |
| | | $\mathcal{L}^{\text{as}}$, (s.in) | 11.15 | 12.50 | 13.68 | 14.24 | 14.46 | 14.57 | 14.83 | 14.93 | 14.98 |
| | | $\mathcal{L}^{\text{as}}$ | 11.18 | 12.54 | 13.71 | 14.24 | 14.46 | 14.59 | 14.84 | 14.94 | 15.00 |
| | $\log D_{\text{KL}}$ | $\mathcal{L}^{\text{sym}}$ | 0.28 | -0.02 | **-0.40** | **-0.66** | **-0.80** | **-0.89** | **-1.12** | **-1.24** | **-1.31** |
| | | $\mathcal{L}^{\text{as}}$, (s.in) | 0.29 | -0.01 | -0.37 | -0.61 | -0.73 | -0.79 | -0.97 | -1.04 | -1.08 |
| | | $\mathcal{L}^{\text{as}}$ | 0.28 | -0.01 | -0.37 | -0.61 | -0.72 | -0.80 | -0.97 | -1.05 | -1.09 |
| CIFAR-10 | OSP | $\mathcal{L}^{\text{sym}}$ | 4.76 | 5.24 | 5.14 | 5.32 | 5.41 | 5.32 | 5.47 | 5.48 | 5.50 |
| | | $\mathcal{L}^{\text{as}}$, (s.in) | 4.90 | 5.29 | 5.33 | 5.47 | 5.49 | 5.56 | 5.54 | 5.57 | 5.57 |
| | | $\mathcal{L}^{\text{as}}$ | 4.76 | 5.07 | 5.51 | 5.56 | 5.55 | 5.57 | 5.60 | 5.59 | 5.62 |
| | $\log D_{\text{KL}}$ | $\mathcal{L}^{\text{sym}}$ | 1.11 | 1.02 | 0.98 | 0.94 | 0.92 | 0.93 | 0.91 | 0.91 | 0.91 |
| | | $\mathcal{L}^{\text{as}}$, (s.in) | 1.09 | 0.99 | 0.93 | 0.90 | 0.88 | 0.87 | 0.86 | 0.86 | 0.86 |
| | | $\mathcal{L}^{\text{as}}$ | 1.10 | 1.01 | 0.91 | 0.88 | 0.88 | 0.86 | 0.85 | 0.85 | 0.85 |

Table 5: Results for the $C_{16}$ experiment with `NEAREST` interpolation. Shown are the metrics measured for different ensemble sizes at the last epoch of training. The numbers presented are means of 30 bootstrapped ensembles of the respective sizes. Bold results that are significantly better (as measured by a $t$-test, $p < .001$) than all other models at the respective metric and size.

