# OpenReview forum: "Ensembles provably learn equivariance through data augmentation"
_ICLR.cc/2025/Conference — Submitted to ICLR 2025_

### Official Review · Reviewer_nmuK · 2024-11-01

**Soundness:** 3
**Presentation:** 3
**Contribution:** 3
**Rating:** 6
**Confidence:** 3

**Summary:**

The paper presents a theoretical analysis showing that data augmentation can lead to equivariance in deep ensembles. The paper's main result is that under several assumptions (e.g. on initialization, architecture, etc.), deep ensembles trained with data augmentation are equivariant in mean, even when individual models are generally not. A similar result was previously presented, but the paper extends these previous results, which were primarily focused on infinitely wide NNs trained with gradient descent under full augmentation, to ensembles of finite-width trained with SGD and random augmentation.
The paper is mainly theoretical and validates the theoretical results through limited and small-scale empirical experiments.

**Strengths:**

1. The paper is well-structured and easy to follow.
1. The paper extends previous results to more reasonable and applicable settings. This is a significant extension.

**Weaknesses:**

I like the paper and believe it has a sufficient contribution and interesting results. However, there are several limitations stated below:

1. While the assumptions for the theoretical analysis are more applicable compared to previous works, they still hold only for infinite-size ensembles. Any analysis (including empirical) on the error bounds for finite ensembles would be beneficial.
1. While the results are important, the novelty is somewhat moderate in the sense that the emergent equivariance property of ensembles was previously proposed and the fact that the theoretical analysis heavily relies on previous works [1].
1. From the empirical evidence, it is unclear if some of the assumptions (like symmetric initialization) are indeed necessary. The authors discuss this, but I believe it can be extended further.
1. Empirical evaluation is limited. It would be beneficial to extend it to more settings, even by small modifications like considering cyclic groups C_k of different orders (k), different architectures, model sizes, etc.
1. It would be beneficial to see the impact of ensemble size on the metrics in Table 1, like adding a line plot for ensemble size vs. OSP. The authors show results for different sizes, but summarizing them in one clear view would make it easier to follow.
1. The paper could benefit from a clearer and more explicit discussion of the limitations of the results.
1. Minor:
    - Line 37: “... a definitive question to the question…”.

Reference

[1] Flinth & Ohlsson, Optimization Dynamics of Equivariant and Augmented Neural Networks, 2023.

**Questions:**

1. Why does the OSP not increase at initialization when ensemble size increases?
1. From the figures, it seems like the results could improve with more epochs (also for baselines). Could you please provide results with a larger number of epochs?

---

> ### Author Response · Authors · 2024-11-19
> **Comments on weaknesses**
>
> We thank for the constructive review.  We are happy to hear that the reviewer thinks that our paper is easy to follow, and that our extension makes the results applicable in more reasonable settings compared to previous results.
>
> All of the points the reviewer makes are valid, as are the suggestions. Let us in the following comment on each on the weaknesses and the questions.
>
> ### Infinite vs finite size ensembles
> It is a reasonable suggestion to include more results about ensembles of finite size. It should be noted that we already have some plots related to the importance of ensemble size in the appendix. We agree that these are somewhat hard to interpret. We have therefore chosen to redo the evaluations, to include more sizes. We have at the time of writing of this rebuttal built new sub-ensembles from our trained models for more ensemble sizes, and measured each of our metrics for the resulting models at epoch 10.
>
> Using a simple t-test on 10 (bootstrapped) samples per size and model, we can confirm with statistical significance (p<.005) that with respect to the KL-divergence,
>
> * $\mathcal{L}^{\mathrm{sym}}$-ensembles are more equivariant than the $\mathcal{L}^{\mathrm{assym}}$ with symmetric initialization for ensemble sizes bigger than or equal to  75 on MNIST, and bigger than 100 on CIFAR.
>
> * $\mathcal{L}^{\mathrm{sym}}$-ensembles are more equivariant than the $\mathcal{L}^{\mathrm{assym}}$ with asymmetric initialization for ensemble sizes bigger than or equal to 25 on MNIST, and bigger than 25 on CIFAR.
>
> See also the following plot (also showcasing OSP) : [Plot](https://anonymous.4open.science/api/repo/ensemble_experiment-1B83/file/graphics/complete_plot.png?v=5df1ae3f)
>
> In the updated version of the paper, we will present data for 30 bootstrapped examples (a setting in which a t-test makes more sense) on all metrics. We can already now conclude that the difference in performance between the different versions is present already for moderate ensemble sizes.
>
> ### Novelty
> We understand and respect the reviewer's point, but hope that they can also agree that the results from the different papers have been put together in a non-trivial way to produce new, meaningful results.
>
> ### Necessary vs. sufficient conditions
> We have indeed only proven sufficient conditions, and we agree that this can be made clearer in the text. We however genuinely believe that proving more than we have already done goes beyond the scope of this work - significantly new ideas need to be applied to obtain a result about convergence towards, rather invariance of, the symmetrical models.
>
> ### More groups
> The reason for only testing the C4 group is that we there have a clean example of where our results apply. When going over to rotation groups of higher order, one starts to need to interpolate, and the invariance condition will not be as clear cut as before.
> We however agree that it is beneficial to also perform experiments in a more 'dirty' setting as far as our theory concerns, since this will provide more information about its practical relevance. We will make one other rounds of experiments, for C16. This will take some time to setup and evaluate, whence we cannot report on results now - we will do this as soon as possible.
>
> ### Limitations
> It is a reasonable suggestion to include a compilation of the limitations in order to increase the readability of the paper. We will do so in an updated version of the paper. As we see it, our main limitations are
> * Our condition is sufficient rather than necessary
> * Our guarantee is only about the infinite-member limit of ensembles.
>
> We can also remark that the following are things we *speculate* on, but *haven't* proven:
> * The extent to which $\Pi_{\mathcal{L}}$ and $\Pi_G$ commute is indicative of emergent equivariance - the smaller it is, the more equivariant the ensembles should be.
> * The set of symmetric models may be an attractor of the dynamics, and not only stable.

---

> ### Author Response · Authors · 2024-11-19
> **Answers to questions**
>
> ### OSP at initialization
> Let us first state that we do not think that this goes against our theory. Instead, we think that this is essentially what is going on: Before training, the predictions of the networks should be more or less random -- that is, the predictions are independent of the data, and rather only are different due to different draws of the parameters at initialization. Thus, the infinite-member ensembles should more or less, for each datum $x$, give one of the 10 classes completely at random. Note that the latter will almost be true also for finite-sample ensembles. Each rotated version hence has a one in ten chance of being the same as the the non-rotation examples, and the expected value of the OSP is $1.3$, which indeed seems to be approximately the OSP of the big ensembles at initialization.
>
> A shorter answer is that this is due to the $\mathrm{argmax}$ function, which is used to determine the predictions, being discontinuous. Note in particular that the KL-divergence-metric is getting smaller when we compare then at 10,100 and 1000 ensemble members (see appendix), so that the ensembles get more and more equivariant at initialization with growing ensemble sizes.
>
> ### Longer training
> We agree that it seems that longer training definitely could lead to more equivariant ensembles. We will however not make any experiments for this, and instead prioritize the C16-experiments. A continuing trend of more and more equivariant ensembles would, as we see it, not say *that* much in this context - the fact that the symmetric ensembles converge faster will still provide the same support to our theory as before. We deem whether the trends continue on another group, where the assumptions are not met in the same clean manner as for C4, a more interesting question, and will therefore prioritize them. We hope the reviewer understands this decision.

---

> > ### Comment · Reviewer_nmuK · 2024-11-25
> > **Response to rebuttal**
> >
> > I would like to thank the authors for their response. I have gone through the reviews and the authors' responses. I believe my main concerns regarding novelty and empirical evaluation remain, and so I will keep my initial score.

---

### Official Review · Reviewer_Ljyp · 2024-11-02

**Soundness:** 3
**Presentation:** 2
**Contribution:** 2
**Rating:** 6
**Confidence:** 3

**Summary:**

The paper expands the results of Gerken & Kessel that show that data augmentation produces equivariant ensembles of models using NTK, by looking at finite network sizes. They then show empirically that their theoretical results indeed hold in practice (up to sampling errors).

**Strengths:**

- It generalizes the results in Gerken & Kessel
- The topic of invariance/equivariance is important so these results would be of interest to people in that community

**Weaknesses:**

My main issue is with the writing:
- The results presented in the main text are quite trivial, that if you start with an invariant distribution and use an invariant flow you end up with an invariant distribution. The more interesting results are in the appendix (appendix B and C)
- You writing $\mathcal{L} = A_\mathcal{L} + T\mathcal{L}$ with $T\mathcal{L}$ the tangent space is very confusing, as tangent space is defined for a manifold and we are talking about a linear space. It needlessly complicates things as there is no need to involve differential geometry when we are working on linear spaces.

**Questions:**

The results in Table 1 aren't that clear to me. In the asymmetric case where you have a symmetric initialization, shouldn't you get results that are similar to the symmetric case? Yet there is a large gap

---

> ### Author Response · Authors · 2024-11-19
>
> We thank the reviewer for their constructive criticism. We are happy to hear that the reviewer finds our results to be of interest to the research community.
>
> ### Presentation of results in main body of text
>
> We agree with reviewer that the main result which is proved in the main text is not as interesting as the result which is proved in Appendix B (the result in the appendix is stronger). Note however that both results are entirely novel, as far as we could tell. Our reasoning for laying out the text as we do is that snce the two proofs follow essentially the same outline, presenting the simpler result in the main text is more pedagogical. That is, the version of our main theorem which could be proved by using the results on equivariant flows from Köhler et al. is presented in the main text precisely *because* it is simpler - less energy is put on the technical details and more on the conceptual.
>
> ### On the notation of the affine space $\mathcal{L}$
>
> The point here is simply that $\mathcal{L}$ is an affine space (linear manifold) and not a linear space (vector space), that is, it can be described as a base point + the tangent space, which in this case is the parallel space going through the origin. Hopefully this clarifies the notation. We could of course choose another terminology for $\mathrm{T}\mathcal{L}$, such as 'parallel space', or the like, but we think that 'tangent space' is the clearest one.
>
> The reason we consider this as the space of linear layers is simply to include more potential architectures into our analysis.
>
> ### The results in Table 1
>
> The results in Table 1 are in line with the theory we have developed. Since the space of convolutions with asymmetric filters (the asymmetric case) is not invariant under the action of the group, our results no longer guarantee the emergence of equivariance, even though they are invariantly distributed at initialization.
>
> It should be noted that the results for the asymmetric model are also quite close to equivariant, which naturally leads to the question if the sufficient condition we have in our theorem is a necessary one or not. In the paper we hypothesize that it may have to do with the fact that the energy of the asymmetric part of the filters is small, so that the asymmetric filters are approximately symmetric in some sense. In Appendix E, we compare what happens in the case of $5\times 5$ filters and we see that the gap between the symmetric and asymmetric indeed grows when the energy of the asymmetric part is increased.

---

> > ### Comment · Reviewer_Ljyp · 2024-11-27
> > **Response**
> >
> > Thank you for your comment. I think the paper has scientific merit, which is why I gave it a score above the acceptance threshold. However, the way the paper is written is important and effects is score.
> >
> > As it is, the main results that make this paper worthwhile are in the supplementary material, not just the proofs but the theorems themselves. This means that the average reader won't even know they exist. Note that as a reviewer "It is not necessary to read supplementary material" making a clear distinction between the main paper and supplementary material.
> >
> > Second, while tangent space might be clearer to some, it requires prior knowledge of differential manifolds. This isn't always the case in the general ML community, as this isn't part of the standard mathematical tools used. As such, adding this without any reason when a simple linear algebra term would suffice is something that I think is problematic as it makes the paper less accessible for no valid reason.

---

> > > ### Author Response · Authors · 2024-11-27
> > >
> > > Thank you again.
> > >
> > > We completely agree that readability of papers is very important. We have changed the word 'tangent space' to 'direction', as used on for example Wikipedia, Planetmath and in Geometric Methods and Applications for computer science and engineering, J. Gallier, Springer, 2011.
> > >
> > > As for the disposition of the text, we completely understand and respect the reviewer's opinion. It would be possible to write the paper only concentrating on the more technically involved version of Theorem 4.2. However, we feel the need to point out that we do not present any theorems in the appendix which are not at least clearly advertized in the main text. Appendix B is only containing lemmas used in the proof of one of the versions of Theorem 4.2, and the proof of that version. Note that the theorem in the main text mentions the case of training with SGD using random augmentation.
> > > We agree that the theorem formulated and proven in appendix C is only mentioned in passing in the main text. However, the statement of the theorem is still there, albeit not in a theorem environment. Note that precisely stating the result is quite involved, and the result is not needed to prove our main result.

---

### Official Review · Reviewer_YfbU · 2024-11-03

**Soundness:** 3
**Presentation:** 3
**Contribution:** 2
**Rating:** 3
**Confidence:** 4

**Summary:**

This paper shows that an ensemble of models when trained with data augmentation leads to emergence of equivariance properties naturally. The results generalize over past known results based on NTKs. The theory assumes some basic assumptions on the architecture and shows that, when the initialization of the weights in an architecture has some symmetry, then, the expected architecture of the ensemble is equivariant. Experimental results with various ensembles validates the results for the C4 group of symmetries.

**Strengths:**

- The work show the emergence of equivariant in ensemble models
- The work generalizes previous works where the proof relied on NTKs
- Experiments with large ensemble of models show the emergence of equivariance

**Weaknesses:**

I have several concerns over the usefulness of the theory and the experimental results.

Usefulness of theory:
- What is the use of the theory in model design or practical use cases? Since equivariant models seems to give perfect equivariance and data augmentation techniques give approximate equivariance. So, I am wondering what is the use of ensemble technique for symmetries, especially, given that we need over 1000 models to get good equivariant results.
- What are the advantages of the proposed technique compared to existing symmetrization and canonicalization methods [1-4] that can convert non-equivariant models into equivariant ones using techniques somewhat similar to ensemble methods but with additional transformations that looks similar to augmentation.

Experimental Results:
- Although the experimental does show that the architecture with symmetric support does give invariant output, but even the asymmetric architecture seems to be giving invariant output, questioning the usefulness of the theory. It is also discussed in the paper about the symmetric states being attractors potentially, but, it still makes the current theory not very useful.
- Experiments are only shown for C4 symmetries

[1] Basu, Sourya, et al. "Equi-tuning: Group equivariant fine-tuning of pretrained models." Proceedings of the AAAI Conference on Artificial Intelligence. Vol. 37. No. 6. 2023.

[2] Mondal, Arnab Kumar, et al. "Equivariant adaptation of large pretrained models." Advances in Neural Information Processing Systems 36 (2023): 50293-50309.

[3] Basu, Sourya, et al. "Efficient equivariant transfer learning from pretrained models." Advances in Neural Information Processing Systems 36 (2024).

[4] Kaba, Sékou-Oumar, et al. "Equivariance with learned canonicalization functions." International Conference on Machine Learning. PMLR, 2023.

**Questions:**

Please see the weaknesses.

---

> ### Author Response · Authors · 2024-11-19
>
> We thank the reviewer for their constructive criticism. We also understand the reviewer's concerns regarding the applicability of the theoretical developments to model design. However, we hope that we can convince the reviewer of the importance of the theoretical developments regardless of their immediate applicability.
>
> The general question that motivated this paper is: "Does data augmentation lead to equivariance?" The technique of data augmentation has been used for a long time in order to align models with various operations, that is, to make them more robust. There is however little in the way of theoretical guarantees of this observed property of data augmentation. In our case we restrict ourselves to studying alignment with symmetries, that is, to emergent group equivariance from data augmentation. In this context, our theoretical results can be viewed as a partial answer to the general question that motivates our research.
>
> ### Usefulness of theory
>
> In our paper, the objective is to show that when training ensembles of networks from scratch under data augmentation, there is an emergent equivariance coming from the optimization process itself. On the other hand, in the papers [1-4], the goal is to develop methods for making a pre-trained model equivariant. In papers [1,3], this is done by averaging the model over the orbit under the group action. This differs from ensembling as considered in out paper, since we average over initializations and random draws of group elements during training. In papers [2,4], it is done by precomposing the model with an equivariant canonicalization map. Although the authors of paper [2] note an augmentation effect of the not yet aligned canonicalization map during training, this is not the cause of the equivariance in this case, and the augmetation effect goes down over time. The results in papers [1-4] are very interesting and we are not suggesting that people should favor our methods over the ones found in these papers. In fact, it is hard to see how our results would apply in the context of *finetuning* foundation models, which is the main focus in at least [1,3]. (They are in principle applicable when the models are trained from scratch).
>
> ### Experimental results
>
> As the reviewer notes, our experimental results seem to indicate that even the models with asymmetric filters become equivariant. This suggests that the sufficient condition in our theorem is not a necessary one. We do not think that this weakens our theory, it merely suggests that further developments are possible. Furthermore, in the paper we hypothesize that this might have to do with the fact that the asymmetric filters are approximately symmetric in the sense that the energy in the asymmetric part is quite small. In Appendix E we provide details on the same experiment performed with $5\times 5$ filters instead of $3\times 3$ filters and we see that indeed the gap between the symmetric and the asymmetric model grows when the energy of the asymmetric part is increased.
>
> ### Experiments beyond C4
>
> The reason for only testing the C4 group is that we there have a clean example of where our results apply. When going over to rotation groups of higher order, one starts to need to interpolate, and the invariance condition will not be as clear cut as before.
> We however agree that it is beneficial to also perform experiments in a more 'dirty' setting as far as our theory concerns, since this will provide more information about its practical relevance. We will make one other rounds of experiments, for C16. This will take some time to setup and evaluate, whence we cannot report on results now - we will do this as soon as possible.

---

> > ### Comment · Reviewer_YfbU · 2024-11-24
> >
> > I thank the author for their response and the additional plots. I have gone through the author's response as well as the other reviews. Unfortunately, my concerns about the usefulness (theoretical/empirical)/non-trivialness of the theory remain. Moreover, the experiments are not convincing enough to make a case for the theory (e.g., the symmetry component important in theory seems to have minimal empirical impact).
> >
> > I look forward to more experiments the authors have promised in their global response. If there is a way to connect the theory and experiments better or provide more use cases (theoretical/empirical), I would be happy to increase my score. But currently, unfortunately, I am unable to do so.

---

### Author Response · Authors · 2024-11-19
**Planned updates**

We would like to thank all the reviewers for their work. Their reviews are all insightful, and contain many valuable suggestion. We have responded to their questions and comments in individual posts.

Let us here only advertize the two big updates we will make to the manuscript before the end of the discussion period.

* We will perform a new set of experiments for the C16 group.

* We will make a more serious evaluation of our models also for smaller ensemble sizes, providing some empirical results in this direction. Here is already a plot formed by measuring the metrics at epoch 10 for the different models (shown is a mean of 10 bootstrapped ensemble samples for each size and model) : [Plot](https://anonymous.4open.science/api/repo/ensemble_experiment-1B83/file/graphics/complete_plot.png?v=5df1ae3f) -- the general trend is that the difference in equivariance is detectable already for moderate ensemble sizes. In the updated version of the paper, we will provide data for 30 bootstraps, and perform some statistical tests. See also the comment to reviewer nmuK.

We will try to make the updates as soon as possible.

---

### Author Response · Authors · 2024-11-25
**Results of updated experiments (I)**

Dear reviewers,

we have had some technical issues, but have now finally managed to run our updated experiments. The results are interesting, and not as clear-cut as one could have wished for. Still, we think that they support the relevance of our theory rather than speak against it.

First, we have, as advertized, re-evaluated our previous experiments (i.e., for $C_4$) for 30 bootstraps instead of 10 bootstraps per sample size. There are no surprises here: the symmetric architecture still outperforms the asymmetric ones, and does so with statistical significance ($p<.001$) from $250$ ensemble members onwards (with respect to the divergence metric, even from $75$ ensemble members). Here is an updated plot: [plot_C4](https://anonymous.4open.science/api/repo/ensemble_experiment-1B83/file/graphics/C4_nearest.png?v=b54e4155)

We have also run the same experiment for the bigger group $C_{16}$. Let us first note that when using this group, we stray from the setting in the paper. The group is no longer acting directly on the support of the images - due to interpolation effects. Hence, the lifted representation $\rho$ on the linear layers $A_i$ no longer perfectly corresponds to rotation of the filters $\varphi_i$ (Example 3.2 is no longer valid). In fact, again due to interpolation, Assumption 1 and 3 are also not given and the spaces $\mathcal{L}$ are no longer invariant.

With this said, here is the plot for our experiments: [plot_C16](https://anonymous.4open.science/api/repo/ensemble_experiment-1B83/file/graphics/C16_nearest.png?v=d3772c9f).

We see that while the symmetric filters still produce more equivariant ensembles than the asymmetric ones on the in-distribution MNIST test data, they are actually not better, and even worse with respect to the divergence metric, on the CIFAR10 data. The most striking difference to the $C_4$ experiments are however that the all of the models are significantly less equivariant on the CIFAR10 data. This was not what was expected. One realizes that this might have to do with the way we have performed our augmentation: We have used the default 'nearest' interpolation option in torchvision to perform the augmentation, and also making sure that the background of the images are uniform. These transformations are not a representation of the group $C_{16}$ -- if we in particular think about filters of size $3\times 3$, the small rotations in fact do nothing.

---

> ### Author Response · Authors · 2024-11-25
> **Results of updated experiments (II)**
>
> We therefore repeated our experiments with the 'bilinear' interpolation option. This is also not a representation of the group in a formal manner, but is at least closer to one -- the action of the small rotations is no longer trivial on small filters, for instance. Here is the plot for those results: [plot_C16_bilinear](https://anonymous.4open.science/api/repo/ensemble_experiment-1B83/file/graphics/C16_bilinear.png?v=23256f1b)
>
> We see that our models now become less invariant on the MNIST data, but more invariant on the CIFAR data. The former can be explained with the fact that the bilinear interpolation will produce images that are blurrier, and also result in a non-uniform background -- the dataset hence becomes more diverse, and it will be a harder problem to learn it. The models can hence not rely on simply learning to perform well on the dataset to become equivariant, as seemed to be enough in the case of using the 'nearest interpolation'.
>
> The still bad performance of the symmetric and asymmetric models is in fact explained by our main result! When rotating an $\mathcal{L}^{\mathrm{sym}}$-filter with $\pi/4$, we will approximately end up with a filter with only non-zero elements on the corners. This is very far from being a $\mathcal{L}^{\mathrm{sym}}$ filter -- the invariance condition is hence far from being satisfied. The asymmetric support for some reason performs slightly better -- we could speculate on why, but ultimately, they perform badly, as would be predicted by the fact that space $\mathcal{L}^{\mathrm{asym}}$ is asymmetric. When repeating the experiments with standard $3\times 3$-filters, something different happens though -- as can be seen in the plot, they vastly outperform the non-standard filters. The corresponding subspace $\mathcal{L}^{\mathrm{cnn}}$ is still not perfectly invariant to non-$\pi/2$-rotations - and also do not yield perfectly equivariant ensembles -- but they are definitely 'more' invariant than both the non-standard filter supports considered -- a rotated $3\times 3$-filter will 'bleed' somewhat, but not as extremely as a $C_4$-symmetric filter.
>
> For full disclosure, we should mention that the bootstraps in the final plot for the non-standard filters are only over approximately 900 total ensemble members -- due to technical difficulties, not all 1000 members finished their training. This will be fixed in a final version. We should course also repeat the CNN experiments also for the nearest interpolation - we will do so in the final version, but already want to report the results we have now for the reviewers to consider.
>
> All in all, we believe that this new set of experiments speaks *in favour* of the practical importance of our theory. Our experiments indicate that in situations where the compatibility condition is not satisfied, the augmentation will *not* lead to equivariant ensembles by itself! One can also note that this aspect of the theory is not at all present in Gerken and Kessel - and only somewhat tangentially in Nordenfors, Ohlsson and Flinth. In this spirit, we thank the reviewers very much for suggesting these experiments.
>
> We understand that we are very close to the end of the discussion period, and understand that the reviewers have already put a lot of effort into reviewing our work, but still hope that they can take the time to consider also these last-minute developments.

---

### Author Response · Authors · 2024-11-27
**Updated version of the pdf**

We have now uploaded an updated version of the pdf. All changes are marked in blue.

In short, the added material is more or less what was posted in our last comment. We have also made some other minor changes, such as changing the terminology "tangent space" and correcting typos.

In a final version of the paper, we will redo the failed experiments for C16 with BILINEAR interpolation, and make an experiment with standard CNN:s for the NEAREST interpolation. We apologize for not being able to do so before the deadline for pdf updates.

We would like to thank the reviewers again their suggested changes. We think that they have improved the paper.

---

### Meta-Review · Area_Chair_h1fU · 2024-12-20

**Metareview:**

This paper studies how equivariance emerges in ensembles of neural networks trained with data augmentation. The authors extend prior theoretical results by showing that equivariance holds under weaker conditions that exist in prior work (e.g. without requiring the NTK limit).
The reviewers appreciated the clear writing and sound theoretical development. However, several concerns were raised about both the theory and experiments: The theoretical contribution it appears to be minor, and the empirical validation was quite limited in scope. The was also some apparent disconnect between the theory predictions and experimental results, which brought into question the relevance of the theory.
The authors engaged constructively with reviewers in the rebuttals.
However, the concerns remain significant enough to prevent me from recommending acceptance.

**Additional Comments On Reviewer Discussion:**

See above.

---

### Decision · Program_Chairs · 2025-01-22

Reject